# BDSB: Brain Disk Schrödinger Bridge for Enhancing 3T BOLD fMRI using Unpaired 7T Data for Visual Retinotopic Decoding

## Abstract

Brain–computer interfaces increasingly rely on retinotopic mapping and visual decoding to reconstruct perceptual experiences from brain activity. High spatial and temporal resolution, coupled with a strong signal-to-noise ratio (SNR), has made 7-Tesla (7T) blood-oxygenation-level-dependent (BOLD) functional magnetic resonance imaging (fMRI) an invaluable tool for understanding how the brain processes visual stimuli. However, the limited availability of 7T MRI systems means that most research relies on 3-Tesla (3T) scans, which offer lower spatial and temporal resolution and SNR. *This naturally raises the question: Can we enhance the spatiotemporal resolution and SNR of 3T BOLD fMRI data to approximate 7T quality?* In this study, we propose a novel framework that aligns 7T and 3T fMRI data from different subjects and datasets in a shared parametric domain. We then apply an unpaired Brain Disk Schrödinger Bridge (BDSB) diffusion model to enhance the spatiotemporal resolution and SNR of the 3T data. Our approach addresses the challenge of limited 7T data by improving the 3T scan quality. We demonstrate its effectiveness by testing it on three distinct public fMRI retinotopy datasets (one 7T, one 3T, and one paired 3T/7T), as well as synthetic data. The results show that our method significantly improves the SNR and goodness-of-fit of the population receptive field (pRF) Kay et al. (2013) retinotopic decoding in the enhanced 3T data, making it comparable to 7T quality. The codes will be available at Github.

## 1 Introduction

Brain–computer interfaces (BCIs) increasingly rely on retinotopic mapping and visual decoding to reconstruct perceptual experiences from brain activity Miyawaki et al. (2008); Du et al. (2022); Sorger & Goebel (2020). Recent studies have demonstrated the feasibility of using retinotopic mapping for BCI control paradigms Chen et al. (2018) and decoding visually imagined content from the visual cortex for communication-oriented BCIs van den Boom et al. (2019). High-resolution functional MRI (fMRI) is central to these efforts, as it provides non-invasive access to the fine-grained organization of the visual cortex Dumoulin & Wandell (2008); Wandell & Winawer (2015).

Beyond BCI applications, researchers have long sought to unravel the mechanisms of visual encoding and decoding through blood-oxygenation-level-dependent (BOLD) signals measured by fMRI. Computational models often analyze voxel-wise time series, from phase-dependent models Engel et al. (1994); DeYoe et al. (1996) used to measure eccentricity, to the widely adopted population receptive field (pRF) framework Dumoulin & Wandell (2008); Kay et al. (2013), and more recent deep learning approaches Thielen et al. (2019). These methods enable delineation of retinotopic maps: a topology-preserving representation of the visual field on the cortical surface Wandell et al. (2007); Xiong et al. (2023). Although its existence has been known for over a century Ribeiro et al. (2024), only recent advances in fMRI have made it possible to map visual areas non-invasively Dumoulin et al. (2003), with growing clinical utility in glaucoma Duncan et al. (2007), Alzheimer's disease Brewer & Barton (2014) and BCIs.

A major challenge in retinotopic mapping is the limited availability of high-quality fMRI scans. Although datasets such as the Human Connectome Project (HCP) Uğurbil et al. (2013); Van Essen

et al. (2013) and the Natural Scenes Dataset (NSD) Allen et al. (2022) use 7-Tesla (7T) fMRI with relatively high resolution and high signal-to-noise ratio (SNR) imaging, their enhanced resolution is not concentrated in the occipital lobe, where retinotopic maps are studied primarily. Still, these scanners provide superior resolution and SNR when compared to more widely available 3-Tesla (3T) machines which were used to generate similar datasets for more general tasks Chang et al. (2019); Horikawa & Kamitani (2017); Gong et al. (2023); Kay et al. (2020). These resources enable investigations of unparalleled scale, resolution, and SNR, empowering researchers to better understand the intricate relationship between visual input and fMRI signals.

Access to high-quality 7T fMRI data would significantly benefit retinotopic mapping and related tasks. For example, standard atlases Wang et al. (2015); Glasser et al. (2016b) used to identify brain areas are often averages in many subjects and are constrained by the resolution of their underlying data. Low resolution can also pose an obstacle to pRF modeling of the time series at each cortical vertex, since the formation of these time series from raw fMRI often involves some spatial smoothing Glasser et al. (2016a). Retinotopic mapping obtained from such pRF modeling may lead to topological violations which disagree with knowledge of cortical physiology and must be corrected before further vision related decoding tasks Tu et al. (2021); Xiong et al. (2023).

With the success of deep learning models in computer vision, researchers have extended these techniques to medical imaging. Generative Adversarial Networks (GANs) Goodfellow et al. (2020) and their variants Armanious et al. (2019); Phan et al. (2023); Zhu et al. (2023), as well as conditional diffusion models Ho et al. (2020); Song et al. (2020); Sasaki et al. (2021); Konz et al. (2024); Dong et al. (2024); Kim et al. (2023); Korotin et al. (2023); Chen et al. (2024a), have shown promising results. However, many of these methods rely on paired data or struggle with unpaired domain alignment, limiting their applicability to certain medical imaging tasks. On the other hand, despite their broad application across various modalities, including fundus imaging Dong et al. (2024); Shen et al. (2020), MRI-CT Li et al. (2020); Zhu et al. (2024); Cui et al. (2024); Huang et al. (2024); Zhang et al. (2024), and fMRI for natural image reconstruction Fang et al. (2020); Ren et al. (2021); Takagi & Nishimoto (2023); Chen et al. (2023; 2024b); Scotti et al. (2024); Wen et al. (2018); Gong et al. (2024), there has been limited focus on enhancing fMRI signals to improve SNR, retinotopic mapping or other neural decoding tasks. This gap highlights the need for methods that specifically target the enhancement of fMRI signals and downstream neural decoding analyses.

To address these limitations, we propose a framework that improves 3T fMRI analyses using unsupervised learning. We map 3D brain surfaces into a shared parametric domain via conformal mapping and apply an unpaired Brain Disk Schrödinger Bridge (BDSB) model to enhance 3T fMRI signals. The resulting fMRI signals preserve cortical structural integrity while approximating the quality and distribution of high-resolution 7T scans, overcoming challenges from short-duration and low-resolution 3T fMRI experiments. Our key contributions are: **(a)** A robust fMRI enhancement pipeline with BDSB model, applied directly to raw fMRI data across different subjects and datasets. **(b)** To our knowledge, it's the first approach to improve fMRI SNR and retinotopic map quality using unpaired learning across public datasets. **(c)** We validate our framework on both real and synthetic experiments, demonstrating its capability to produce high-quality fMRI scans and improve downstream visual-related neural decoding tasks such as retinotopic mapping and pRF analysis.

## 2 METHODOLOGY

### 2.1 DATASETS AND EXPERIMENTAL DESIGNS

Fig. 1 shows our pipeline. We start with three datasets: the 7T *Natural Scenes Dataset* (NSD) Allen et al. (2022), the 3T *Natural Object Dataset* (NOD) Gong et al. (2023), the 3T/7T *Temporal Decomposition Method for task-based fMRI* (TDM) Kay et al. (2020). NSD contains approximately 40 sessions per subject for 8 participants, including natural images and pRF-fLoc stimuli Benson et al. (2018); Stigliani et al. (2015), providing high-quality (HQ), fine-resolution fMRI data. In contrast, NOD includes 10 to 63 sessions per subject for 30 participants, with 9 subjects performing pRF-fLoc tasks and the rest viewing only images, offering broader but lower-quality (LQ) 3T scans in similar experiments. Additionally, TDM includes 11 subjects with 4 task-based fMRI experiments, among which 2 subjects ($s_1, s_3$) underwent one eccentricity stimuli session in both 3T and 7T resolution, offering limited but comparable data with paired LQ and HQ fMRI for identical subjects.

Figure 1: Overview of the Pipeline: We design three different experimental designs from real NOD, NSD, TDM data and synthetic NSD data. All the experiments go through our pipeline with three components described in Sec. 2.

Ideally, we seek to evaluate our pipeline using datasets in which the same subjects were scanned at both 3T and 7T resolution under identical visual stimuli, but publicly available datasets meeting these criteria are extremely limited. Among them, only TDM partially fulfills this requirement, providing paired 3T and 7T data for 2 subjects who underwent eccentricity-based visual stimulation. Unfortunately, each subject has only a single session for this task, making the dataset too small to support large-scale training or subject-agnostic modeling. To overcome this limitation, we design two main experiments in addition to using TDM alone. First, to enable evaluation with known ground truth, we construct a synthetic dataset by down-sampling HQ 7T data to simulate LQ 3T-like inputs. This allows us to directly compare model outputs with the original 7T data. Second, we assess the model's generalization and cross-dataset performance, where LQ and HQ fMRI are drawn from different subjects and datasets. In this case, the real 7T fMRI for input subjects are unknown, so we train the model to map real 3T data toward the distribution of 7T data using unpaired examples, thus enabling enhancement of 3T signals without requiring subjects to undergo costly 7T scanning.

Details of the three experimental designs are: **(a) Synthetic Data**[1]: The original NSD fMRI provides HQ targets, and their down-sampled versions act as LQ inputs. To simulate LQ fMRI, we use Neuromaps Markello et al. (2022) to transform all pRF sessions from the 164k fsaverage surface to the 32k fsLR surface, matching the space resolution of the 3T data like NOD. We add Gaussian noise to each vertex in the transformed fMRI time series to simulate signal degradation. This process creates synthetic LQ data with corresponding HQ counterparts, enabling ground-truth evaluations. The first 6 NSD subjects are used for training and the remaining 2 subjects are reserved for testing. **(b) Cross-Dataset Real Data**: All 8 NSD subjects serve as HQ targets, while the first 7 NOD subjects with pRF tasks act as LQ sources during training with the remaining 2 NOD subjects reserved for testing. Since we do not have ground truth 7T fMRI for NOD subjects, we can only evaluate the results by the overall Fréchet inception distance (FID) and the downstream pRF decoding performance. **(c) TDM Real Data**[1]: We use the only 2 subjects from TDM who took the eccentricity stimuli. Due to the small number of subjects, we use the first 3 runs out of 6 for train and the last 3 runs for test. A summary of the experimental strategies is shown in Tab. 1.

Table 1: Our experimental strategies. For each subject $s_a$ from the NSD or NOD datasets, all trials involving the pRF stimuli are included. For the TDM dataset, only $s_1$ and $s_3$ underwent both 7T and 3T experiments; thus, we include the trials from their single eccentricity stimuli session.

| Experiments | Train and Valid | | Test and Output | |
|---|---|---|---|---|
| | Source | Target | Source | Target |
| Synthetic[1] | Down-sampled NSD $s_1 \sim s_6$ | 7T NSD $s_1 \sim s_6$ | Down-sampled NSD $s_7, s_8$ | 7T NSD $s_7, s_8$ |
| Cross-Dataset Real | 3T NOD $s_1 \sim s_7$ | 7T NSD $s_1 \sim s_8$ | 3T NOD $s_8 \sim s_9$ | no ground truth |
| TDM Real[1] | 3T TDM $s_1, s_3$ runs $1 \sim 3$ | 7T TDM $s_1, s_3$ runs $1 \sim 3$ | 3T TDM $s_1, s_3$ runs $4 \sim 6$ | 7T TDM $s_1, s_3$ runs $4 \sim 6$ |

---

[1]Even when paired LQ/HQ data is available for subjects, training is performed in an unpaired manner—i.e., the target fMRI corresponds to a randomly selected subject $s_b$, not the same subject as the input subject $s_a$.

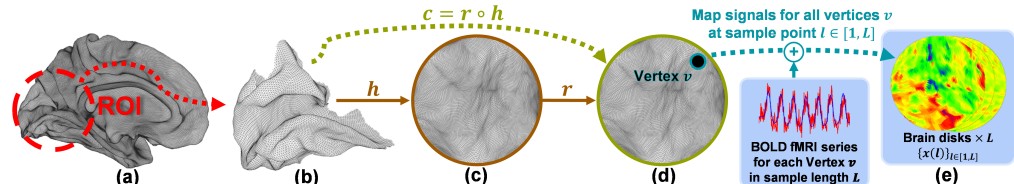

Figure 2: Disk conformal parameterization: (a) full fsaverage surface mesh; (b) ROI subdivision of the full mesh from FreeSurfer vertex labels Fischl (2012); (c) the parameterized planar disk of the ROI obtained from harmonic map $h$; (d) the refined planar disk $\boldsymbol{x}'$ through resulting conformal mapping $c = r \circ h$; (e) the BDs $\{\boldsymbol{x}(l)\}_{l \in [1,L]}$ generated by mapping BOLD fMRI time-series $\{y_v(l)\}_{l \in [1,L]}$ for each vertex $v$ to their corresponding locations on the refined planar disk $\boldsymbol{x}'$.

## 2.2 BRAIN DISK PARAMETERIZATION

To translate LQ fMRI slices into HQ counterparts, we need to align the probability spaces of 3T/7T trials across datasets. Given the SNR variations and structural differences across subjects, a shared domain is necessary. We achieve this by using the 164k fsaverage Fischl (2012) cortical surface and conformal mapping to generate parameterized brain disks for our region of interest (ROI).

**Shared Surface Mesh and ROI.** We start with the original 3D surface meshes from each dataset. The NSD provides native surface meshes (approximately 220k vertices per hemisphere) for each subject, along with transformation files in FreeSurfer format Allen et al. (2022); Fischl (2012). Using the NSD code packages Allen et al. (2022), we retrieve the 3D coordinates of faces and vertices, along with vertex annotations, and transform them into the 164k fsaverage surface. In contrast, the NOD dataset stores its surface meshes in the 32k fsLR format using Ciftify Dickie et al. (2019). The surface data and vertex annotations are retrieved via Ciftify toolbox Dickie et al. (2019). These meshes are then transformed into the 164k fsaverage surface using Neuromaps toolbox Markello et al. (2022) with linear approximation. For TDM subjects, we directly use the native surface meshes (approximately 200k vertices) which are already aligned across 3T and 7T sessions Kay et al. (2020).

To streamline analysis, we define a ROI encompassing vertices labeled as *lateraloccipital*, *cuneus*, *pericalcarine*, and *lingual*, representing key cortical regions in the occipital lobe. This ROI ensures inclusion of most primary visual cortex while significantly reducing computational overhead.

**Conformal Mapping.** In order to train our enhancement model in 2D space instead of the 3D surface meshes, we choose the widely used conformal parameterization Tu et al. (2021); Ta et al. (2022); Xiong et al. (2023; 2024) $c : M \to D$, to map a cortical surface mesh $M$ to a unit disk $D$. Given $M$ as an open boundary genus-0 surface after cutting to ROI, the harmonic map $h : M \to D'$ minimizes the energy Jin et al. (2018); Gu et al. (2004): $E(h) = \int_M |\nabla h|^2 \, dv_M$. For disk-like surfaces, the harmonic map $h$ satisfies the Laplace equation Wang et al. (2007): $\Delta h(u)_M = 0, \quad h|_{\partial M} = g$, where $\Delta$ denotes the Laplacian operator and $g : \partial M \to \partial D'$ is a boundary mapping given by arc-length parameterization. In discrete cases, the harmonic map is efficiently obtained by solving the sparse linear system $L_h h = 0$, where $L_h$ is the Laplacian matrix Wang et al. (2007).

Denote a disk to disk refinement $r : D' \to D$, the final conformal mapping would be a composition of $c = r \circ h$. To achieve conformality for mapping $c$, we can refine $r$ by iteratively modifying the mapping until its Beltrami coefficient $\mu_r$ satisfies $||\mu_r||_\infty \leq \epsilon_{\mu_r}$ Wang et al. (2007); Ta et al. (2022).

The final parameterization $c = r \circ h$ produces 2D Brain Disks (BDs), where each vertex's fMRI signal is conformally projected onto the disk, ensuring spatially consistency across subjects and datasets. Fig. 2 illustrates the full process, with BDs visualized by showing BOLD fMRI signal values in RGB color representation. Finally, for a single trial fMRI series of $L$ samples, we can produce $L$ BD slices: $\{\boldsymbol{x}(l)\}_{l \in [1,L]}$.

## 2.3 BRAIN DISK SCHRÖDINGER BRIDGE ENHANCEMENT

**Background.** The Schrödinger Bridge Problem (SBP) finds the optimal stochastic process $\{x_t : t \in [0,1]\}$ that transforms an initial distribution $p_0$ into a target distribution $p_1$. Formally, the SBP is

Figure 3: Illustration of BDSB and all loss terms. For a randomly selected time step $t_i \in \mathbf{t}$, we recursively generate samples following Eq. 2 and the joint distribution to approximate distribution $\hat{x}_{1|t_i} \sim p(x_1, x_{t_i})$ as discussed in Sec. 2.3. All losses and regulations ($\mathbb{L}_{\text{Adv}}, \mathbb{L}_{\text{SB}}, \mathbb{L}_{\text{reg}}$) are combined with individual weights described in B.1.

defined as $T^\star = \arg\min_{T \in \mathcal{Q}(p_0, p_1)} D_{\text{KL}}(T \| W^\tau)$, where $W^\tau$ is the Wiener measure with variance $\tau$, and $\mathcal{Q}(p_0, p_1) \subset \mathcal{P}(\Omega)$ is a stochastic process that requires the boundary distributions (i.e., start at $p_0$ and end at $p_1$). The solution $T^\star$ is called the Schrödinger Bridge (SB), which connects the boundary distributions $(p_0, p_1)$ and works as an optimized result for the entire trajectory. In our application, $p_0$ and $p_1$ represent 3T and 7T BD distributions viewing the same pRF stimuli, and $T^\star$ provides a probabilistic path bridging them. The target is to generate enhanced high-quality BDs $\{\hat{\boldsymbol{x}}_1(l)\}$ from input low-quality BDs $\{\boldsymbol{x}_0(l)\}$.

**Discrete Bridge Approximation.** The continuous SBP can be approximated via a sequence of Entropic Optimal Transport (EOT) problems over successive time intervals $[t_a, t_b] \subseteq [0, 1]$ Dong et al. (2024); Kim et al. (2023); Korotin et al. (2023); De Bortoli et al. (2021):

$$T^\star_{t_a, t_b} = \arg\min_{\gamma \in \Pi(T_{t_a}, T_{t_b})} \mathbb{E}_{(x_{t_a}, x_{t_b}) \sim \gamma} \|x_{t_a} - x_{t_b}\|^2 - 2\tau(t_b - t_a)H(\gamma), \tag{1}$$

where $\gamma \in \Pi(T_{t_a}, T_{t_b})$ represents all possible joint marginal distributions consistent with the boundary states $T_{t_a}$ and $T_{t_b}$ (i.e., the most likely distributions of BDs over arbitrary interval), and $H(\cdot)$ is the entropy function. Additionally, for $t \in [t_a, t_b]$, denote $s(t) = (t - t_a)/(t_b - t_a)$, the conditional distribution $p(x_t \mid x_{t_a}, x_{t_b})$ follows the Gaussian distribution Tong et al. (2023):

$$p(x_t \mid x_{t_a}, x_{t_b}) \sim \mathcal{N}\Big(s(t)x_{t_b} + (1 - s(t))x_{t_a}, \ s(t)(1 - s(t))\tau(t_b - t_a)\mathbf{I}\Big), \tag{2}$$

By fixing $t_b = 1$ and discretizing the interval as $\boldsymbol{t} := \{t_i\}_{i=0}^N$, we can compute $T^\star_{t_i, 1}$ sequentially. The joint distribution $p(x_1, x_{t_i}) = p(x_1 \mid x_{t_i})\, p(x_{t_i})$ with $p(x_{t_i}) = p(x_0) \prod_{j=0}^{i-1} p(x_{t_{j+1}} \mid x_{t_j})$ can be iteratively approximated using Eq. 2 under the Markov assumption. Consequently, this procedure yields the SB trajectory to iteratively approximate the desired distribution $p(x_1, x_{t_i})$ as Fig. 3.

**BDSB Learning.** It's necessary to obtain the posterior $p(x_1 \mid x_{t_i})$ in order to compute the objective function in Eq. 1. We use a neural generator $q_\phi(x_1 \mid x_{t_i})$ with $(x_{t_i}, t_i)$ as inputs and $\phi$ as parameters. The SB objective function over the sub-interval $[t_i, 1]$ is then reformulated as:

$$\min_\phi \mathbb{L}_{\text{SB}}(\phi, t_i) := \mathbb{E}_{q_\phi(x_{t_i}, x_1)} \|x_{t_i} - x_1\|^2 - 2\tau(1 - t_i)H(q_\phi(x_{t_i}, x_1))$$

$$\text{subject to:} \quad \mathbb{L}_{\text{Adv}}(\phi, t_i) := D_{\text{KL}}(q_\phi(x_1) \| p(x_1)) = 0 \tag{3}$$

where $q_\phi(x_{t_i}, x_1) := q_\phi(x_1 \mid x_{t_i})p(x_{t_i})$, and the constraint ensures that the generator learns the high-quality distributions. By introducing a Lagrange multiplier, Eq. 3 can be reformulated into

$$\min_\phi \mathbb{L}_1(\phi, t_i) := \mathbb{L}_{\text{Adv}}(\phi, t_i) + \lambda_{\text{SB}}\mathbb{L}_{\text{SB}}(\phi, t_i) \tag{4}$$

Solving Eq. 4 with the optimal parameters $\phi$ achieves $q_\phi(x_1 \mid x_{t_i}) = p(x_1 \mid x_{t_i})$ and $q_\phi(x_{t_i}, x_1) = p(x_{t_i}, x_1)$ for every steps $t_i$ Kim et al. (2023); Dong et al. (2024). Consequently, the generator $q_\phi(x_1 \mid x_{t_i})$ can be directly utilized to sample the next BD $x_{t_{i+1}}$ for every steps $i = 0, 1, \ldots, N-1$. Through the iterative process shown in Fig. 3, we can generate the final enhanced fMRI response $x_{t_N}$ (i.e., the enhanced $\hat{x}_1$) starting from the initial 3T distribution $x_0 \sim p_0$. The training and inference details are outlined in B.1.

However, optimizing $\mathbb{L}_1$ alone does not guarantee $\hat{x}_1$ to preserve the structural details of the brain since $\mathbb{L}_1$ only ensures the optimal transformation path between signals (i.e., enhancing fMRI values

Table 2: Metrics on enhanced fMRI and down-stream pRF results for all three experiments.

| Experiments | Metrics | raw LQ | Cycle-GAN | OTT-GAN | OTE-GAN | SCR-Net | fast-DDPM | Proposed |
|---|---|---|---|---|---|---|---|---|
| Synthetic | SSIM ↑ | 0.475 | 0.760 | 0.803 | 0.783 | 0.525 | 0.566 | **0.855** |
| | PSNR ↑ | 14.24 | 22.98 | 23.39 | 22.16 | 14.64 | 15.26 | **25.05** |
| | FID ↓ | 152.3 | 126.7 | 72.70 | 77.41 | 108.6 | 71.40 | **42.88** |
| | $\bar{R}^2$ ↑ | 18.30 | 17.22 | 18.01 | 16.89 | 13.54 | 15.53 | **24.00** |
| Cross-Dataset Real | FID ↓ | 183.83 | 139.69 | 96.90 | 95.91 | 177.8 | No pair data | **70.65** |
| | $\bar{R}^2$ ↑ | 20.26 | 19.78 | 19.99 | 18.64 | 15.11 | No pair data | **25.91** |
| TDM Real | SSIM ↑ | 0.402 | 0.602 | **0.727** | 0.702 | 0.454 | 0.511 | 0.718 |
| | PSNR ↑ | 13.00 | 18.70 | 19.18 | 19.06 | 13.79 | 14.06 | **19.24** |
| | FID ↓ | 166.7 | 134.9 | 84.45 | 88.00 | 107.1 | 96.91 | **62.09** |

but distort BD structure). Here, we incorporate two regularization terms: **(a)** PatchNCE Dong et al. (2024); Kim et al. (2023) between enhanced $\hat{x}_1$ and its low-quality counterpart $x_0$. **(b)** Brain disk structural similarity measure (BD-SSIM) between the generated BDs and the original fsaverage BD structure $x'$. The final loss function $\mathbb{L}_2$ is now defined as:

$$\mathbb{L}_2(\phi, t_i) := \mathbb{L}_{\text{Adv}}(\phi, t_i) + \lambda_{\text{SB}}\mathbb{L}_{\text{SB}}(\phi, t_i) + \sum_{l=\text{nce,bd}} \lambda_{\text{Reg}_l}\mathbb{L}_{\text{Reg}_l}(\phi, t_i) \tag{5}$$

## 2.4 RE-SAMPLING AND pRF DECODING

With a well-trained BDSB, we can generate enhanced versions of 3T fMRI BDs. Due to the bijective nature of conformal mapping, the fMRI response for every vertex $v$ at each sample point $l$ can be re-sampled from corresponding pixel $\boldsymbol{x}_v$ on BDs. By aggregating enhanced BDs across all sample points $\{\hat{x}_1(l)\}$ within a single pRF trial, we reconstruct a complete enhanced fMRI $\boldsymbol{y} = \{y_v(l)\}$.

We employ pRF decoding as a downstream neural decoding task to quantify the improvements. Given a vertex-wise fMRI signal series $\boldsymbol{y} = \{y_v(l)\}$, the pRF model Dumoulin & Wandell (2008); Kay et al. (2013); Waz et al. (2024) predicts the receptive center $\boldsymbol{c}_v = (c_v^{(1)}, c_v^{(2)})$ and size $\sigma_v$ on the visual field. A predicted fMRI signal for vertex $v$ is given by:

$$\hat{y}_v(\boldsymbol{c}_v, \sigma_v, l) = \beta \left[ \int_{\boldsymbol{z} \in \text{visual field}} r(\boldsymbol{z}; \boldsymbol{c}_v, \sigma_v) s(l, \boldsymbol{z}) \, d\boldsymbol{z} \right] * h(l) \tag{6}$$

where $h(l)$ is the hemodynamic response, $r(\boldsymbol{z}; \boldsymbol{c}, \sigma)$ is a Gaussian kernel, and $s(l, \boldsymbol{z})$ is the visual stimuli. The parameters $\{(\boldsymbol{c}_v, \sigma_v)\}$ are estimated by minimizing the prediction error:

$$(\boldsymbol{c}_v, \sigma_v) = \arg \min_{\boldsymbol{c}_v, \sigma_v} \sum_{l \in [1, L]} \|\hat{y}_v(\boldsymbol{v}, \sigma, l) - y_v(l)\|^2 \tag{7}$$

The pRF results are obtained by solving Eq. 7 for every vertices on the cortical surface. The quality of fit is usually evaluated using variance explained $R_v^2$ for each vertex $v$ (percentage): $R_v^2 := (1 - \frac{\sum_{l \in [1, L]}(\hat{y}_v(l) - y_v(l))^2}{\sum_{l \in [1, L]}(y_v(l) - \bar{y}_v)^2}) \times 100\%$ Dumoulin & Wandell (2008); Kay et al. (2013); Waz et al. (2024). Higher $R_v^2$ indicates the signal of $v$ align closer to the visual stimuli with better interpretability.

## 3 EXPERIMENTS AND RESULTS

**Hyperparameter and Training**. The generator and discriminator follow the architectures outlined in Kim et al. (2023); Dong et al. (2024), with further training details listed in B.1.

**Evaluation Metrics.** We assess enhancement quality using standard similarity metrics (SSIM, PSNR, FID) and downstream *q-pRF* analysis Kay et al. (2013); Waz et al. (2024). For TDM dataset, only similarity metrics are reported due to their simplified stimuli. More details are in B.1.

**Enhanced fMRI Results.** We adopt five 2D translation models to our pipeline as baselines (details of baseline models can be found in supplementary material). Quantitative results are summarized in

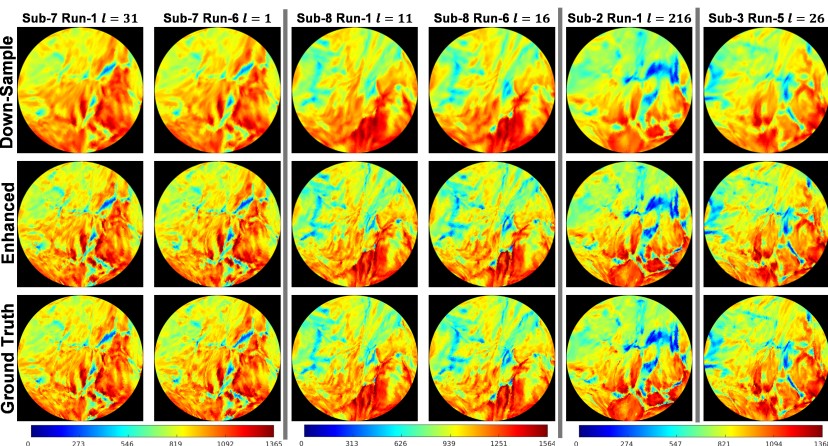

Figure 4: Illustration of down-sampled LQ, enhanced, and original HQ BDs across different NSD subjects, trial runs, and sample points. fMRI values are shown as RGB colors at their vertex location on the parameterized planar disk mapped from the 3D fsaverage surface. See More results in B.2.

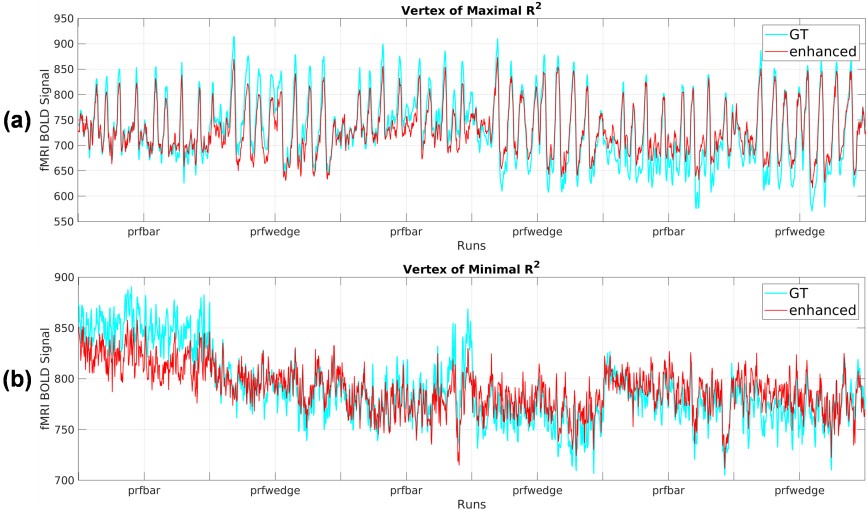

Figure 5: BOLD signals from the HQ ground truth, synthetic LQ, and the enhancement. (a) The enhanced fMRI signal for the vertex with highest $R^2$ shows a significant alignment to the ground truth, with slight misalignment on valley points. (b) The vertex with minimal $R^2$ shows a worse alignment, likely due to its minimal visual response and inactive signal pattern on low $R^2$ vertices. See more results in B.2

Tab. 2. Across all real and synthetic experiments, our pipeline achieves the best performance, significantly enhancing LQ fMRI signals to better approximate HQ scans and improving pRF analysis. In contrast, baseline models generate spurious BDs to increase similarity but distort brain surface structures, leading to poorer performance in fMRI retrieval and much worse pRF results which is indicated by average $R^2$ confidence across the ROI.

We illustrate the enhanced fMRI on the same parametric brain disks obtained via conformal mapping in Fig. 4. The enhanced disks display finer spatial resolution and a more distinct fMRI distribution that aligns closely with the underlying cortical structures, particularly in regions with extreme values or high curvature. To further analyze the performance, we compared the ground truth and the enhanced BOLD time series for two distinct vertices within our ROI: one with a strong response to the pRF stimuli and another with minimal response (determined by the value $R^2$ from the pRF decoding). As shown in Fig. 5, the enhanced signal closely matches the ground truth for the active vertex, demonstrating a strong performance in capturing extreme values. However, for inert vertices,

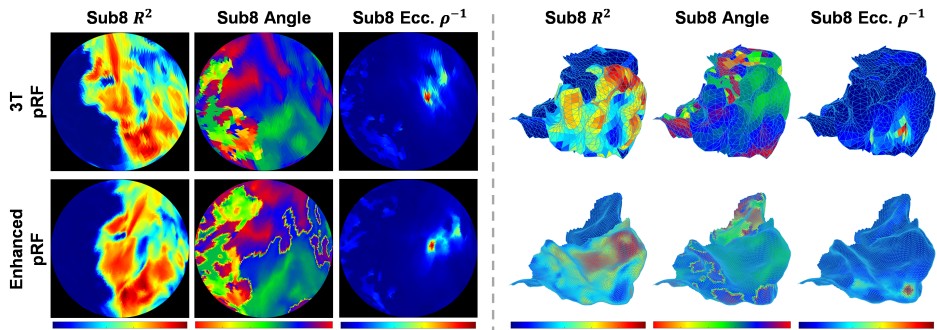

Figure 6: Illustration of pRF results from the NOD test subject $s_8$ on both the 2D parametric domain and the 3D brain ROI mesh. Our enhancement preserves brain structure through consistent receptive fields (note that different vertex labels between fsLR and fsaverage may cause slight ROI shifts), while achieving much higher retinotopic mapping resolution and $R^2$ confidence across most ROI regions. Additional visualization results are provided in the B.2.

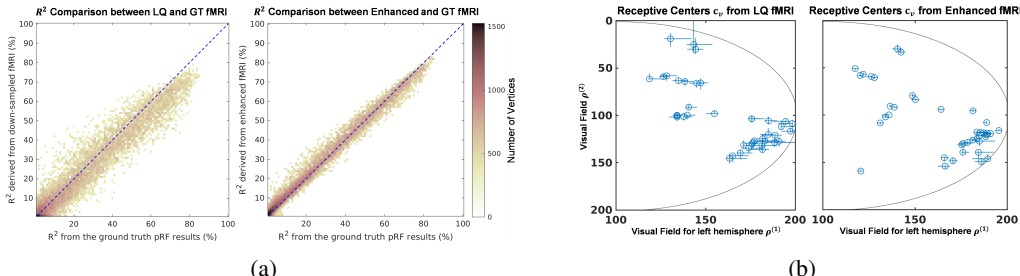

Figure 7: **(a)** $R^2$ values from down-sampled NSD fMRI (left) show greater variability and lower overall confidence compared to HQ data, while enhanced fMRI (right) significantly improves $R^2$ consistency, especially among high-confidence vertices. **(b)** Receptive centers $c_v$ decoded from vertices with top-40 $R^2$ values under random sampling. LQ fMRI (left) exhibits unstable and dispersed estimates, whereas enhanced fMRI (right) yields more stable and consistent localization.

where the signal remains relatively constant, the alignment is weaker. This discrepancy likely arises from the difficulty in learning unchanging values based on their distribution across the brain disks.

**Enhanced pRF Results.** In addition to evaluating enhanced fMRI signals directly, we assess downstream performance to demonstrate the broader utility of our pipeline. Fig. 6 compares pRF results from the NOD fMRI and those derived from enhanced fMRI time series on the parametric domain and 3D mesh. The enhanced signals yield notably higher $R^2$ values across many vertices on a much higher surface resolution, while preserving spatial organization in consistent receptive fields, despite the absence of paired supervised training.

For synthetic data experiment, we further validate the performance by comparing the $R^2$ values of the native 7T pRF parameters with those derived from down-sampled and enhanced fMRI, as shown in Fig. 7(a). The down-sampled fMRI exhibits greater variance and generally lower $R^2$ values compared to the ground truth high-quality pRF results. In contrast, the $R^2$ values derived from the enhanced fMRI data shows significant improvement, particularly for vertices with high $R^2$, while maintaining a comparable confidence threshold to the ground truth.

To evaluate temporal stability, Fig. 7(b) shows receptive centers $c_v$ of the top-40 highest $R^2$ left-hemisphere vertices across 50 independent pRF analyses using random stimulus intervals. Unlike typical pRF models trained on merged time series from all session runs, randomized intervals will examine the variability and interpretability of the fMRI time series for different stimuli segments.

Our results indicate that enhanced fMRI signals yield lower variability and more consistent receptive centers across randomized intervals of the pRF stimuli. These results confirm that our enhancement

pipeline preserves the interpretability and reliability of pRF decoding, bridging the gap to 7T data in downstream retinotopic decoding tasks.

Table 3: Ablation study of our framework on: (a) different brain mapping strategies, including direct slicing, harmonic mapping, and conformal mapping; (b) the effect of different regularization terms in BDSB. All experiments are conducted on the synthetic experimental setting.

| Brain Mapping | $Reg_{nce}$ | $Reg_{bd\text{-}ssim}$ | SSIM | PSNR | FID | $\bar{R}^2$ |
|---|---|---|---|---|---|---|
| Slice | ✗ | ✗ | 0.237 | 8.24 | 226.8 | 6.102 |
| Harmonic $h$ | ✗ | ✗ | 0.833 | 24.19 | 35.56 | 16.97 |
| Conformal $c$ | ✗ | ✗ | 0.849 | 24.26 | **34.23** | 22.02 |
| Conformal $c$ | ✓ | ✗ | **0.858** | 24.88 | 42.64 | 21.88 |
| Conformal $c$ | ✓ | ✓ | 0.855 | **25.05** | 42.88 | **24.00** |

**Ablation Study**. Tab. 3 presents an ablation analysis evaluating the contribution of each component in our pipeline. Unlike MRI-based models, direct slicing of the cortical surface introduces geometric distortions in the disk representation, leading to inconsistent training data and terrible performance. Harmonic mapping also reduces effectiveness by failing to preserve face areas in the transformation from 3D meshes to 2D parametric disks, which impairs spatial fidelity. Regarding regularization, PatchNCE loss provides modest gains by encouraging similarity between enhanced and input signals. In contrast, BD-SSIM loss plays a critical role in maintaining structural integrity of the brain disk, leading to notable improvements in both BOLD signal quality (measured by PSNR) and functional decoding accuracy (reflected by average $\bar{R}^2$ in pRF analysis).

# 4 CONCLUSION AND DISCUSSION

We present a robust fMRI enhancement pipeline that maps BOLD signals from 3D cortical surfaces onto 2D parametric brain disks. Using an unpaired Brain Disk Schrödinger Bridge diffusion model, our method enhances 3T fMRI signals by learning from unpaired 7T data. It achieves signal quality and downstream performance comparable to native 7T scans, while preserving both functional responses and cortical geometry. This enables more accurate pRF modeling and supports a range of vision-related fMRI decoding tasks.

**Lack of Paired Data.** A central challenge in this emerging area is the absence of large-scale paired 3T–7T visual fMRI datasets. Unlike structural imaging (e.g., MRI or CT), there are almost no standardized resources where the same subjects are scanned at both field strengths under identical visual stimuli. We therefore adopt an unpaired learning framework and benchmark against both supervised and unsupervised baselines to ensure fairness. To partially mitigate this gap, we also incorporate the only available paired resource (TDM), though its scope is limited to two subjects and non-standard stimuli. Our experiments highlight both the difficulty and the importance of this limitation, underscoring the community need for future standardized, paired datasets.

**Synthetic Data.** Given the scarcity of paired data, we emphasize synthetic and cross-dataset evaluation. While down-sampling and noise injection provide a principled proxy for low-quality signals, such synthetic 3T-like data cannot fully capture scanner hardware, pulse sequence, or subject-level variability. We therefore complement synthetic benchmarks with real-data experiments, balancing control with realism. This dual strategy directly addresses such concerns by demonstrating both methodological validity and practical robustness, while calling for more realistic simulation protocols and shared benchmarks. More discussion continued in B.3.

**Future Work.** Although our current focus is on pRF analysis and retinotopic decoding, the proposed framework is broadly applicable to many downstream tasks which require fMRI quality from low field strengths, such as fMRI-based segmentation, classification, and visual reconstruction. A promising direction is to extend our approach from ROI-aggregated samples to vertex-level fMRI time series, enabling more precise modeling of localized cortical dynamics. With continued development, our method has the potential to set a new standard for improving 3T or 1.5T fMRI quality, effectively narrowing the gap toward 7T-level resolution and expanding the capabilities of functional neural decoding in both research and clinical applications.

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

# A APPENDIX

## A.1 ETHICS STATEMENT

This research fully complies with the ICLR Code of Ethics https://iclr.cc/public/CodeOfEthics. Our work uses only publicly available, anonymized fMRI datasets Allen et al. (2022); Gong et al. (2023); Kay et al. (2020); Benson et al. (2018), all of which were collected with informed consent and ethical approvals by their original providers. No new data involving human or animal subjects were collected for this study, and therefore no additional institutional review board (IRB) approval was required.

We emphasize that our contributions respect all ethical terms set forth by ICLR: - **Research conduct.** The methodology, experiments, and results are presented with transparency, and we plan to release code and processed data to enable reproducibility. - **Respect for human subjects.** All data are de-identified and provided by prior studies with appropriate consent and oversight; our analysis does not attempt to re-identify subjects or access any private information. - **Privacy and fairness.** Since the datasets are public and anonymized, there are no risks of violating privacy or introducing unfair treatment of individuals or groups. - **Dual use and misuse.** Our method is intended for scientific research to enhance low-field fMRI quality and downstream neuroscience analyses. While any generative technique may carry hypothetical risks of misuse, we believe this risk is minimal here, as our framework operates solely on brain imaging data and produces outputs that remain within the scope of academic research.

By adhering to these principles, we ensure that our work is conducted responsibly, legally, and with due consideration of both potential benefits and limitations.

## A.2 REPRODUCIBILITY STATEMENT

We have taken several steps to ensure that our work is fully reproducible. All datasets used in this study are publicly available and properly cited, including the 7T Natural Scenes Dataset (NSD) Allen et al. (2022), the 3T Natural Object Dataset (NOD) Gong et al. (2023), and the 3T/7T Temporal Decomposition Method dataset (TDM) Kay et al. (2020). Details of how these datasets were accessed, preprocessed, and incorporated into our experiments are described in the methodology sections 2 and B.1. All toolboxes and code packages employed (e.g., FreeSurfer, Ciftify, Neuromaps) are explicitly listed and cited in the methodology sections and supplementary material. To support reproducibility, we provide complete descriptions of our experimental designs, hyperparameters, and evaluation protocols in Sec. 3 and supplementary materials. Finally, we will release our full implementation, including training code, model checkpoints, and preprocessing pipeline, upon publication of this work, enabling independent verification and extension of our results.

## A.3 USE OF LARGE LANGUAGE MODELS

In this work, large language models (LLMs) were used solely as an assistive tool for language editing and polishing of the text (e.g., grammar correction, clarity improvements, and stylistic refinement). No part of the research process—including problem formulation, dataset selection, methodological

design, model implementation, experiment execution, analysis, or result interpretation—relied on LLMs. The scientific contributions, technical content, and experimental results are entirely the work of the authors. We take full responsibility for the accuracy and integrity of the manuscript.

# B SUPPLEMENTARY MATERIALS

## B.1 ADDITIONAL EXPERIMENTAL DETAILS

### B.1.1 DATA SPLITS AND EVALUATION DETAILS

To clarify the construction of our three experimental settings, we provide detailed descriptions of the dataset splits. Across all settings, training is conducted in an **unpaired** manner—i.e., the low-quality (3T) inputs and high-quality (7T) targets always come from different subjects (or even different datasets) unless otherwise noted. This ensures that our method does not rely on trivial voxel- or subject-level correspondences, but instead learns to align distributions across domains.

- **Synthetic Data Experiment.** We use the 7T NSD dataset as the source of high-quality (HQ) fMRI. Synthetic low-quality (LQ) data are generated by down-sampling from 164k fsaverage to 32k fsLR surfaces and injecting Gaussian noise. Subjects 1–6 are used for training and validation, while subjects 7–8 are held out for testing. During training, enhanced fMRI from some subjects are generated with reference to HQ fMRI from randomly selected other subjects.

  During testing, we have: down-sampled fMRI, the generated enhancement fMRI, and the original ground-truth HQ fMRI. This design provides direct ground-truth for evaluating SSIM and PSNR. FID is always applicable regardless of ground-truth availability.

  Since these sessions are from standard pRF stimuli, we can also run downstream pRF decoding.

- **Cross-Dataset Real Data Experiment.** We use the 3T NOD dataset as the LQ source and the 7T NSD dataset as the HQ target. Training is unpaired: NOD subjects 1–7 serve as inputs, while all NSD subjects (1–8) serve as targets. NOD subjects 8–9 are held out for testing. During training, enhanced fMRI from NOD subjects are generated with reference to HQ fMRI from random NSD subjects.

  During testing, we have: real LQ fMRI from NOD, the generated enhancement fMRI, and real HQ fMRI from NSD for target reference (but no paired ground truth). Thus, only FID can be computed directly.

  Since both datasets are from standard pRF stimuli, we can run downstream pRF decoding to further assess functional improvements.

- **TDM Real Data Experiment.** We use the two available subjects (s1, s3) who were scanned with eccentricity stimuli at both 3T and 7T. Runs 1–3 of the 3T scans are used as input and runs 1–3 of the 7T scans as targets, while runs 4–6 are reserved for evaluation. During training, enhanced fMRI from 3T recordings of s1 or s3 are generated with reference to randomly chosen 7T recordings from random subjects (i.e., when enhance s1, the HQ reference may be s1 or s3).

  During testing, we have: real LQ fMRI, the generated enhancement fMRI, and the original paired HQ fMRI. This allows direct measurement of enhancement quality using SSIM and PSNR. FID is also computed as it does not require pairing.

  Since the TDM dataset used simplified eccentricity stimuli rather than full pRF stimuli, downstream pRF decoding cannot be performed.

Taken together, these three complementary experiments provide a solid and comprehensive evaluation. The synthetic setting ensures quantitative ground-truth benchmarking, the cross-dataset real setting demonstrates generalization to unseen subjects and datasets without paired supervision, and the TDM setting offers the only available partial pairing for visual fMRI. By combining all three, our evaluation addresses the lack of large-scale paired data while still validating the robustness, interpretability, and generality of our approach.

### B.1.2 BASELINE DETAILS

In order to ensure fairness, all baseline models and our proposed Brain Disk Schrödinger Bridge (BDSB) are trained and tested on the same data split. They also share the same preprocessing of fMRI BDs for generation, and the same quality assessment pipeline from qPRF Waz et al. (2024) for downstream evaluation.

Implementation details for each model are provided below:

**Cycle-GAN** Zhu et al. (2017). The model was trained for 200 epochs using the RMSprop optimizer, with initial learning rates of $0.5 \times 10^{-4}$ for the generator and $1 \times 10^{-4}$ for the discriminator. A linear decay schedule was applied, reducing the learning rate by a factor of 10 every 100 epochs. Training used a batch size of 4, with input BDs resized to $256 \times 256$ and augmented via random horizontal and vertical flips. The final objective combined the GAN loss, cycle consistency loss, and identity loss with weights $\lambda_{GAN} = 1$, $\lambda_{Cycle} = 10$, and $\lambda_{Idt} = 5$, respectively. The GAN loss was computed using Mean Squared Error (MSE), while the cycle and identity losses were based on the L1-norm.

**OTT-GAN** Zhu et al. (2023). This model adopted the same generator and discriminator architectures as Cycle-GAN, following the designs in Zhu et al. (2023). In addition to the standard objectives, an optimal transport (OT) loss was included, with a weighting factor of $\lambda_{OT} = 40$. The OT loss was computed using MSE to better align data distributions across different BD surfaces.

**OTE-GAN** Zhu et al. (2023). Similar to OTT-GAN, the OTE-GAN model employed the same baseline architecture and training strategy. The primary difference was in the computation of the OT loss, which was based on the MS-SSIM metric instead of MSE, emphasizing perceptual similarity rather than pixel-wise accuracy.

**SCR-Net** Li et al. (2022). The model was trained for 150 epochs using the Adam optimizer, with an initial learning rate of $2 \times 10^{-4}$ and $\beta_1 = 0.5$, followed by an additional 50 epochs during which the learning rate was linearly decayed to zero. The training batch size was set to 16. All BDs were resized to $256 \times 256$ and augmented via random flipping. The generator and discriminator architectures followed the configurations described in Li et al. (2022).

**Fast-DDPM** Jiang et al. (2024). The model was trained using 10 diffusion steps—slightly more than the 5 steps used by BDSB. All input BDs were resized to $256 \times 256$. Training was conducted using the Adam optimizer with $\beta_1 = 0.9$, a learning rate of $2 \times 10^{-5}$, and no weight decay. The batch size was set to 16, and the model was trained for 400 epochs. Other hyperparameters and architectural details followed the default settings in the official implementation Jiang et al. (2024).

### B.1.3 LOSS TERM DETAILS

For completeness, we describe in detail the three loss terms used in our BDSB framework, as illustrated in Fig. 3 of the main paper. The key idea is that each loss term regularizes the enhancement process from different perspectives: fidelity to HQ signals, preservation of LQ input content, and maintenance of structural integrity of the BDs.

- **Adversarial Loss ($\mathbb{L}_{\mathbf{Adv}}$).** This loss is computed as the divergence between the distribution of predicted BDs with enhanced signals $\hat{x}_1$ and the referencing HQ BD $x_1$. It ensures that the generated outputs are indistinguishable from real HQ data, enforcing global distribution alignment. Without this adversarial component, the generator may produce enhancements that match individual metrics but fail to capture realistic overall signal statistics.

- **Generation Loss ($\mathbb{L}_{\mathbf{SB}}$).** This loss is computed during each generation steps. It ensures that the generated output distribution matches the statistics of HQ data and previous step, thereby guiding the model to recover high-quality fMRI patterns. This corresponds to the Schrödinger Bridge objective in Eq. 3 of the main text.

- **NCE Regularization Loss ($\mathbb{L}_{\mathbf{Reg}_{nce}}$).** This loss is computed between the predicted BD with enhanced signals $\hat{x}_1$ and the input BD with LQ signals $x_0$, using a contrastive (PatchNCE) formulation Kim et al. (2023). It constrains the enhancement not to drift too far from the input distribution, thereby preserving subject-specific responses and preventing mode collapse.

- **BD-SSIM Loss ($\mathbb{L}_{\mathbf{Reg}_{bd-ssim}}$).** This is a novel structural regularization term, computed between the predicted BD with enhanced signals $\hat{x}_1$ and the structural BD $x'$ (i.e., BD without fMRI values, simply the mapped ROI surface structure) that encodes cortical geometry without fMRI signals. Unlike conventional SSIM-based losses (which compare output to input or to HQ references), BD-SSIM explicitly enforces the generated enhancement to align with the underlying cortical structure of the parameterized brain disk. This unique formulation ensures that enhanced signals respect anatomical consistency, reducing distortions observed in baseline models. Our ablation study shows that BD-SSIM substantially improves both signal fidelity (PSNR) and downstream functional decoding accuracy ($\bar{R}^2$).

The final training objective is the weighted combination shown in Eq. 5. In summary, the adversarial loss enforces distribution realism, the generation loss ensures fidelity to HQ signals, the NCE regulation loss maintains consistency with LQ inputs, and the BD-SSIM regulation loss uniquely preserves cortical structure. Together, these complementary terms provide both quantitative improvements and functional interpretability.

### B.1.4 EXPERIMENTAL DETAILS

As stated in the main paper, key components in BDSB—namely, the generator and Markovian discriminators—follow the architectures outlined in Kim et al. (2023); Dong et al. (2024). For the Schrödinger Bridge diffusion process, we discretize the continuous unit interval $[0, 1]$ into $N = 5$ time steps. The loss function includes weights $\lambda_{SB} = 1$, $\lambda_{\mathrm{Reg}_{nce}} = 0.5$, and $\lambda_{\mathrm{Reg}_{bd-ssim}} = 1$, corresponding to the Schrödinger Bridge loss, PatchNCE regularization Kim et al. (2023); Dong et al. (2024), and BD-SSIM regularization, respectively. We also set $\pi = 0.01$.

For all three experimental settings, despite differing dataset sizes, all input BDs were resized to $256 \times 256$, and the batch size was set to 8. Models were trained for 150 epochs using the Adam optimizer with $\beta_1 = 0.5$, $\beta_2 = 0.999$, and an initial learning rate of $1 \times 10^{-4}$, which decayed linearly after 75 epochs. Training was conducted on a single *NVIDIA GeForce GTX TITAN X*, with total training time approximately 45, 41, and 5 GPU hours for the NSD synthetic, NSD-NOD cross-dataset, and TDM dataset experiments, respectively.

During testing and generation, the batch size was set to 1. A typical fMRI run of around 300 seconds required less than 2 minutes to generate the corresponding enhanced signal for the entire ROI, covering approximately 14,000 vertices per brain hemisphere.

### B.1.5 PRF TASK DETAILS

Before being fed into the population receptive field (pRF) analysis, each input fMRI time series is detrended on a per-stimuli basis. Specifically, the entire pRF experiment consists of separate sessions, and within each session, a second-order (quadratic) polynomial is fitted and removed from the signal using least-squares projection. This effectively eliminates low-frequency drifts and scanner-related trends while preserving task-relevant fluctuations. The result is a set of detrended fMRI signals that are temporally normalized and well-suited for accurate pRF estimation.

We used the qPRF package Waz et al. (2024) to perform downstream pRF estimation. For a typical NSD subject undergoing 6 sessions of 300-frame pRF stimuli, or a NOD subject with 8–12 sessions of 150-frame stimuli, qPRF required approximately 100 seconds to estimate optimal pRF parameters across the full ROI (about 14,000 vertices per hemisphere).

Thanks to the roughly $1000\times$ speed-up offered by qPRF compared to conventional pRF methods Kay et al. (2013), we were able to efficiently iterate on our method using downstream task performance as feedback.

## B.2 ADDITIONAL RESULTS

### B.2.1 ENHANCED FMRI SIGNAL EVALUATION

Due to space constraints in the main paper, we present additional visualizations of enhanced fMRI signals here, highlighting both BD and temporal perspectives. These examples include results from

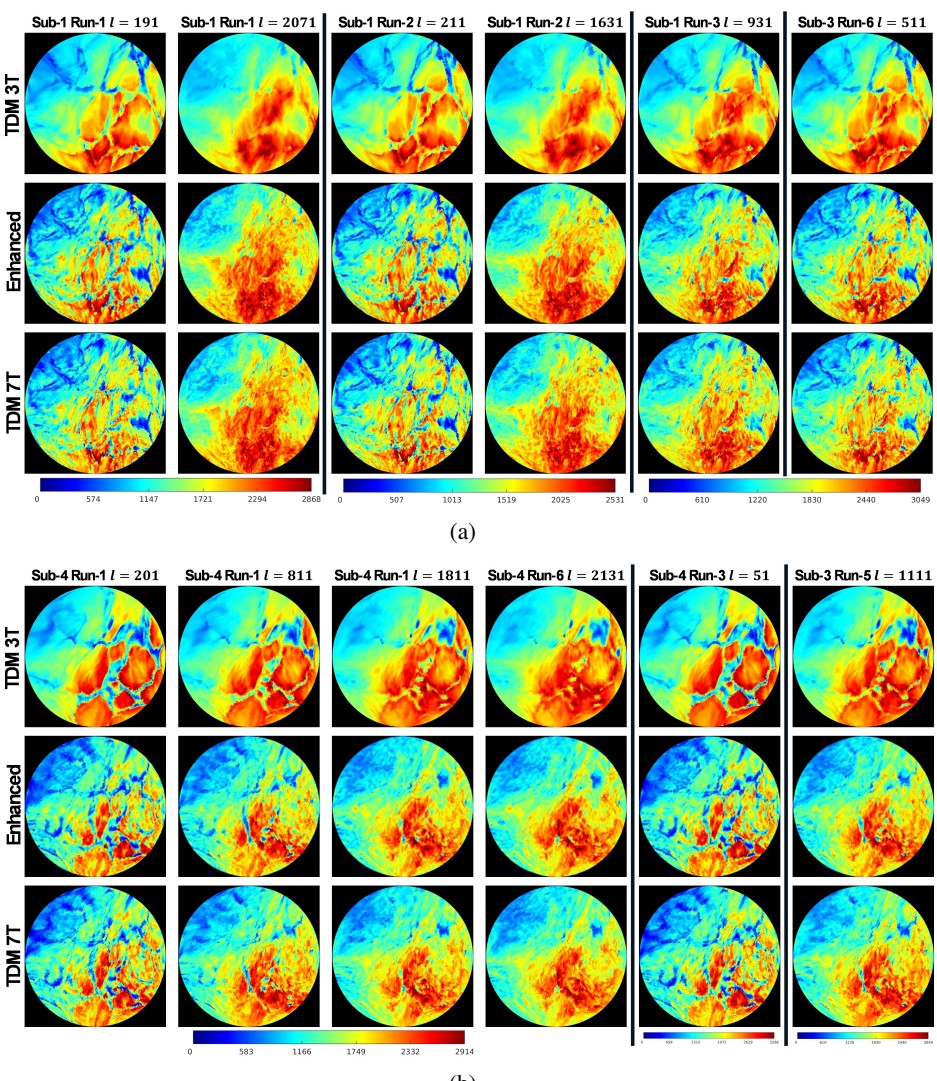

Figure 8: Visualization of original 3T, enhanced, and original 7T fMRI signals on brain disks (BDs) across selected sessions and sample points from TDM subjects (a) $s_1$ and (b) $s_3$. Although the TDM dataset contains only eccentricity-based stimuli, it is the only known dataset providing matched 3T and 7T acquisitions for the same subjects. Beyond the synthetic enhancement results shown for NSD, our pipeline also performs reliably on real-world 3T/7T data without requiring paired supervision. All fMRI values are rendered as RGB colors based on vertex-level values on the 2D parameterized disk derived from the 3D fsaverage surface.

the TDM real data and NSD synthetic data experiments, demonstrating the effectiveness and robustness of our proposed pipeline in two distinct settings: enhancing real-world 3T-to-7T fMRI data and recovering high-quality 7T-like signals from degraded synthetic low-quality inputs.

**BD Visualization.** As illustrated in Fig. 8, the BD representations of enhanced signals reveal clear improvements over low-quality (LQ) inputs, closely approximating high-quality (HQ) 7T signals. Despite the TDM dataset's limitations—specifically, the use of only eccentricity-based stimuli (which precludes standard pRF analysis) and its restricted subject pool (only subjects $s_1$ and $s_3$ underwent both 3T and 7T scans)—it remains, to our knowledge, the only publicly available dataset featuring 3T/7T pairs from the same individuals. Our model demonstrates consistently strong enhancement performance on this real-world dataset, comparable to the synthetic experiments on NSD

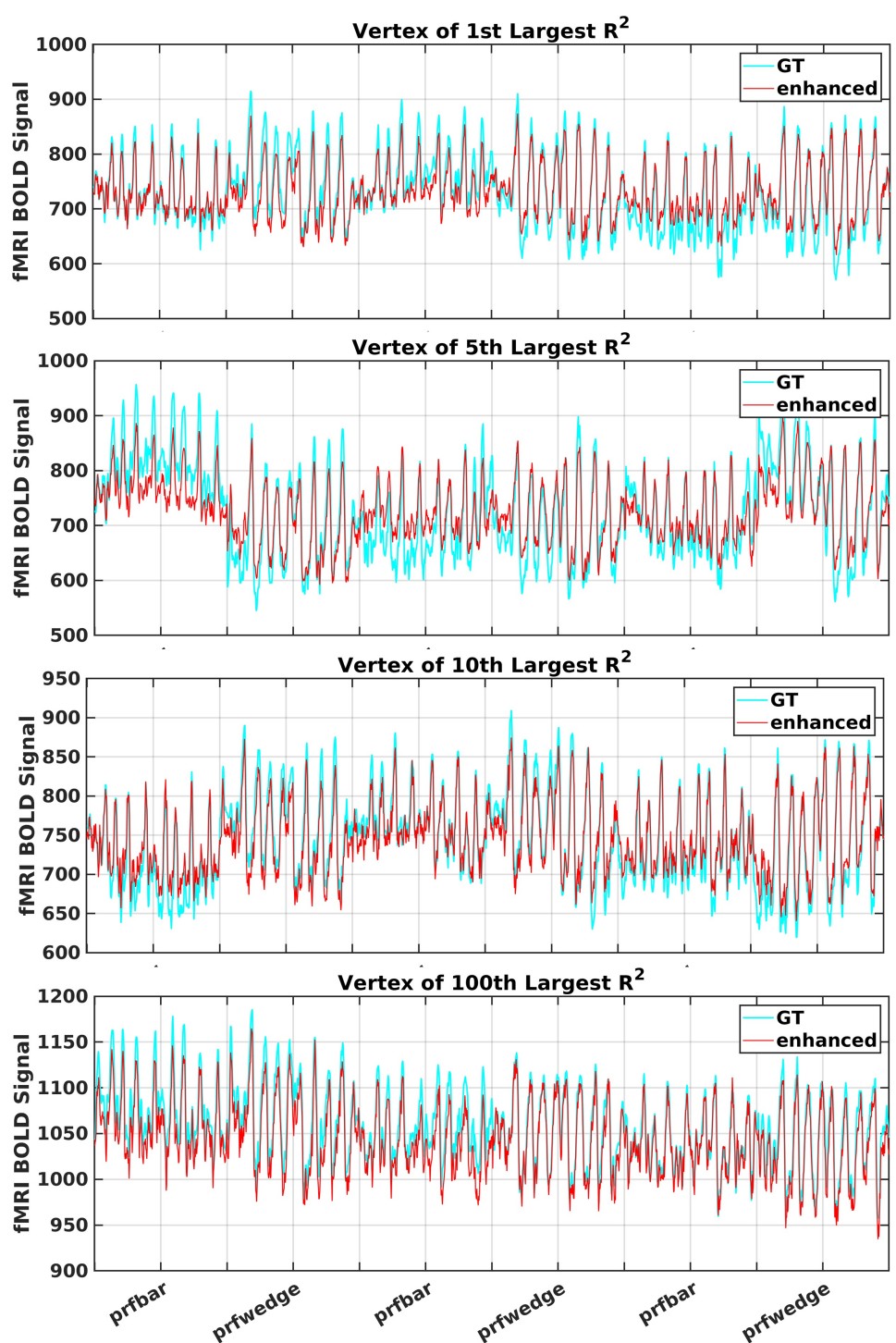

Figure 9: Temporal BOLD signal comparisons between the original 7T fMRI and enhanced outputs, focusing on vertices with high $R^2$ values. The enhanced signals closely track the ground truth, especially for top-ranking vertices. Minor deviations are observed around troughs and rapid transitions, but the overall dynamics remain well-preserved. Alignment quality gradually decreases as $R^2$ lowers, consistent with reduced visual responsiveness in those regions.

data. These results affirm the pipeline's generalizability and potential for deployment in practical and clinical neuroscience scenarios.

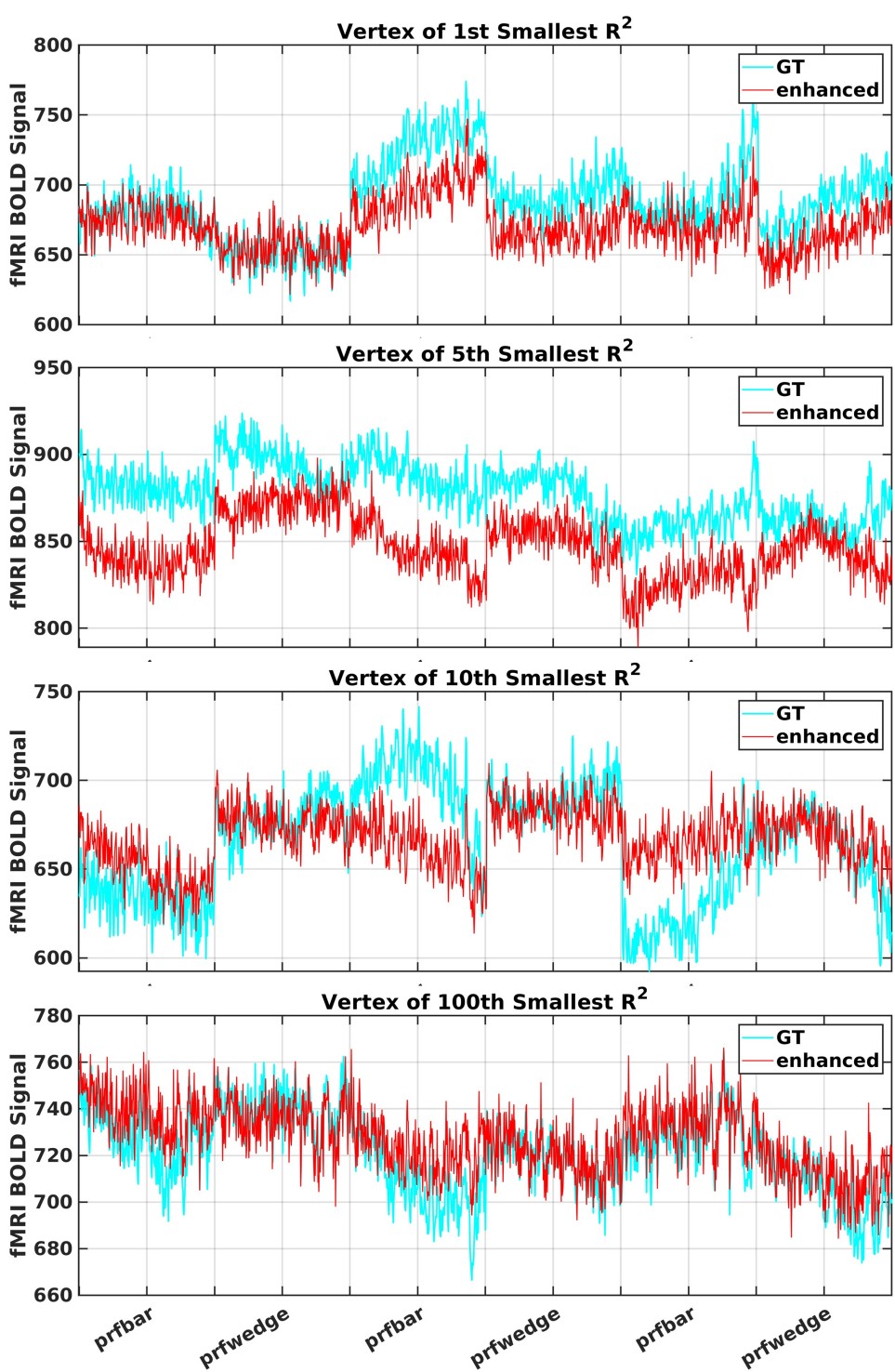

Figure 10: Temporal BOLD signal comparisons between the original 7T fMRI and enhanced outputs, focusing on vertices with low $R^2$ values. The enhancement results show reduced alignment due to minimal visual activation and the dominance of noise in these regions. As $R^2$ increases, alignment quality improves steadily, indicating the model's capacity to selectively enhance meaningful task-related components.

**Time Series fMRI Visualize.** In Fig. 9 and Fig. 10, we present temporal BOLD signal traces from selected cortical vertices ranked by their $R^2$ values—specifically, the top-1, top-5, top-10, and top-

100 highest and lowest. These values are derived from the quality of the pRF model fits. The enhanced fMRI signals show strong temporal alignment with the original high-quality 7T signals, particularly for vertices with high $R^2$ scores. This indicates the model's ability to recover biologically meaningful and visually evoked patterns.

In contrast, vertices with low $R^2$ scores exhibit weak or absent stimulus-driven responses, often dominated by noise or non-task-related fluctuations. These patterns are inherently harder to model and predict, which explains the decreased alignment observed in low $R^2$ regions. Nevertheless, our approach still maintains a coherent temporal structure in the enhanced signals, without introducing artificial oscillations or phase shifts.

### B.2.2 ENHANCED PRF RESULTS

We provide additional pRF mapping results from the unsupervised cross-dataset experiments between NOD and NSD. The pRF maps derived from both the original low-quality 3T fMRI and our enhanced fMRI are visualized on the 2D parametric surface and the corresponding 3D cortical meshes. Results are shown for two training subjects ($s_1$, $s_2$) and one unseen test subject ($s_9$). The results for another held-out test subject ($s_8$) are presented in the main paper.

Fig. 11 highlights the effectiveness of our proposed pipeline in improving the quality of 3T fMRI signals. The enhanced outputs yield consistently higher average $R^2$ values, sharper and more anatomically coherent region boundaries, and improved eccentricity topographies. Crucially, the enhanced signals enable robust pRF estimation even on high-resolution cortical meshes, while faithfully preserving each subject's individual visual response patterns.

In the absence of ground truth 7T data or pRF solutions for the NOD dataset, our enhanced signals and the corresponding pRF estimates serve as strong approximations of high-field data. This demonstrates the potential of our method to provide reliable and clinically useful reconstructions for subjects who can only undergo low-field scans—a particularly important consideration in real-world and clinical neuroscience settings.

### B.2.3 SIGNIFICANCE TEST

To validate the reliability and statistical robustness of our method, we conduct significance testing based on 5-fold cross-validation experiments on the NSD synthetic experiment (i.e., 5 different combinations of 6 subjects for training, and the remaining 2 for testing). For each fold, we compute the performance metrics including the average pRF confidence $\bar{R}^2$ across the ROI region and the SSIM score between enhanced fMRI and the original fMRI. The performance metrics remain consistent across folds, with our method achieving an average $\bar{R}^2$ of $24.00 \pm 2.41$ and SSIM of $0.855 \pm 0.034$, compared to the best baseline OTT-GAN Zhu et al. (2023) performance of $18.09 \pm 3.15$ and $0.803 \pm 0.020$, respectively.

To assess statistical significance, we apply a paired t-test between the fold-wise metrics of our method and the best baseline OTT-GAN. The resulting p-values are $p = 0.004$ for the NSD synthetic experiment, indicating that the improvements achieved by our method are statistically significant at the 0.01 level. These results confirm that the observed gains are not due to random variation from choice of subjects, and our enhancement pipeline consistently outperforms existing approaches across different training strategies.

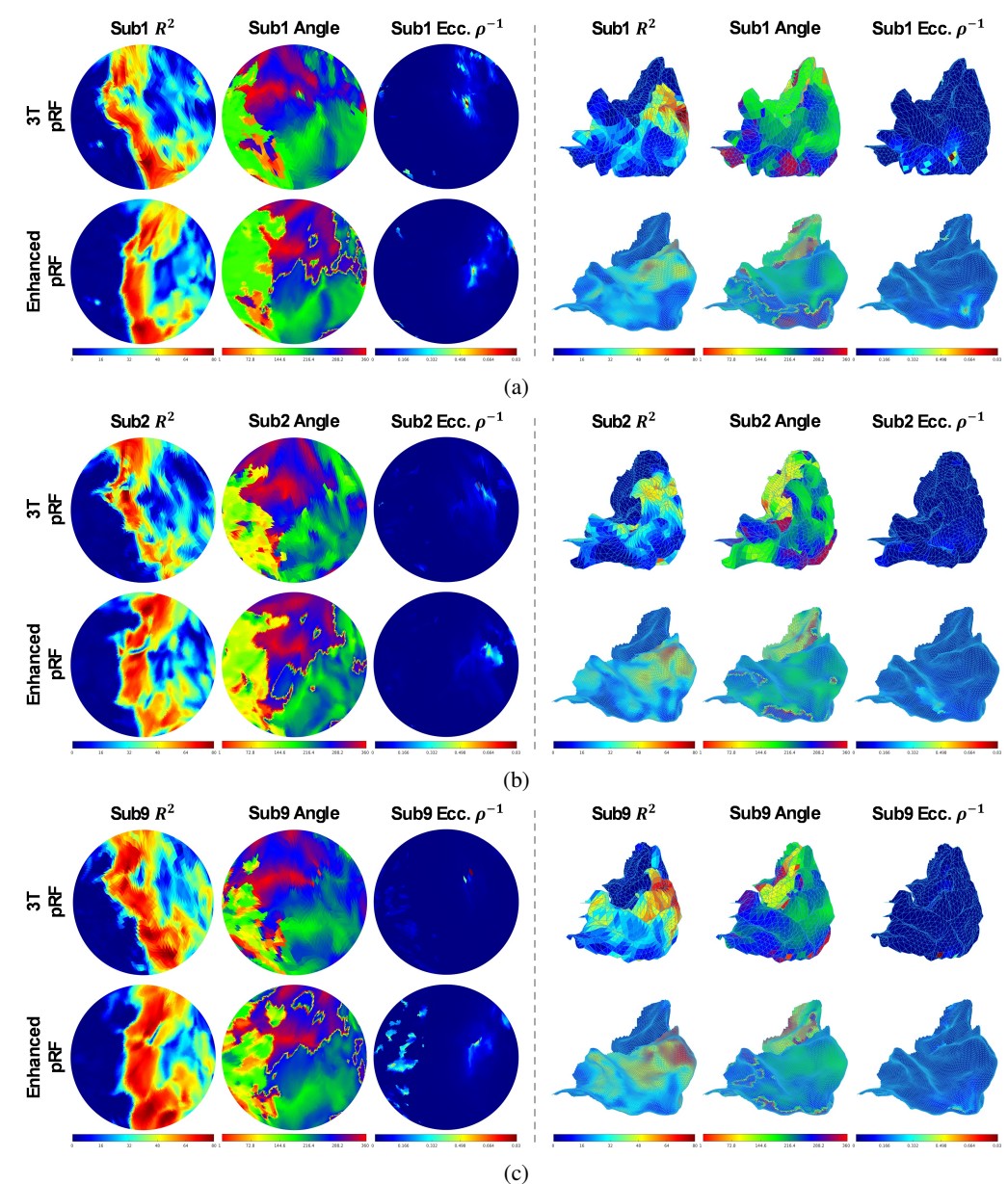

Figure 11: Visualization of pRF results from NOD subjects: (a) trained subject $s_1$, (b) trained subject $s_2$, (c) unseen test subject $s_9$. Each panel shows the original and enhanced pRF maps on both the 2D parametric surface and 3D cortical mesh. Despite the lack of high-resolution supervision, the enhanced results exhibit clear retinotopic structures and significantly improved $R^2$ distributions across ROI regions. Notably, consistent receptive field layouts are preserved, with minor ROI alignment variations due to registration and surface label differences (e.g., between fsLR and fsaverage spaces).

### B.3 FURTHER DISCUSSIONS

**Can we enhance temporal structure?** Our model enhances each fMRI time frame independently, yet it is trained across the entire sequence of frames from pRF experiments (300 frames for 7T, 150 for 3T). This ensures that consistent spatial features are learned and temporal patterns are implicitly preserved. Comparisons between ground truth and enhanced signals Fig. 5 and supplementary

Fig. 9,10 confirm that stimulus-driven fluctuations are retained. Furthermore, improved pRF $R^2$ scores demonstrate that temporal fidelity is maintained in downstream decoding.

**ROI selection and computation cost.** We restrict analysis to visual cortex regions—lateral occipital, cuneus, pericalcarine, and lingual cortices—based on the aparc label file in the fsaverage surface. This ensures alignment across datasets and minimizes confounds. While this focus matches the datasets' emphasis on retinotopy, the pipeline can be extended to other brain regions when appropriate task-driven datasets are available. It is possible to work the pipeline on a larger ROI for different tasks (such as language related fMRI using other ROI), but expanding to larger ROIs requires a higher-resolution BD, increasing computation and memory requirements. For our current work focusing on retinotopic mapping, the current ROI and BD resolution fit each other.

**Real enhancement or hallucination?** A key question is whether enhancement preserves neural responses rather than hallucinating plausible patterns. Our evaluation with ground-truth data (synthetic and TDM) demonstrates close alignment between enhanced and true signals. Moreover, receptive field estimates (Fig. 6) show consistent cortical topologies. These analyses indicate that enhancement preserves meaningful neural patterns while reducing noise.

**Mapping distortions.** Conformal mapping introduces some spatial distortion. However, because both source and target BDs are aligned to fsaverage and BD-SSIM regularization is applied, distortions are minimized and back-projection to cortical vertices remains valid. Our measurements confirm that distortions are acceptable for reliable fMRI reconstruction.

**How's the improvement in downstream $\bar{R}^2$ scale?** It is important to interpret the magnitude of $R^2$ values in context. With our chosen ROI ( 14k vertices across visual cortex), the average $\bar{R}^2$ is typically around 20% for raw 3T data and 25% for raw 7T data (When restricting to a more localized regions, such as V1–V3 ( 1k vertices), the average $\bar{R}^2$ for 7T data can reach  60%). On the current ROI scale. our enhanced data increase the pRF performance close to 7T level (Tab. 2). Moreover, it increases the number of vertices exceeding $R^2 > 70\%$ (Fig. 7a), showing that improvements are meaningful relative to accepted scales according to retinotopic mapping literature Kay et al. (2013); Dumoulin & Wandell (2008).

