# OpenReview forum: "BDSB: Brain Disk Schrödinger Bridge for Enhancing 3T BOLD fMRI using Unpaired 7T Data for Visual Retinotopic Decoding"
_ICLR.cc/2026/Conference — Submitted to ICLR 2026_

### Official Review · Reviewer_YvSq · 2025-10-25

**Soundness:** 3
**Presentation:** 2
**Contribution:** 3
**Rating:** 6
**Confidence:** 3

**Summary:**

To address limited availability of 7T data, this paper proposes a novel method for enhancing resolution and SNR of 3T BOLD fMRI to approximate 7T data quality. It aligns both 3T and 7T data from different subjects and datasets into the same space with shared parameters. By using an unpaired Brain Disk Schrodinger Bridge (BDSB) diffusion model, the spatio-temporal resolution and SNR of 3T data could be enhanced. Experiments were performed on three public fMRI datasets and a synthetic dataset. The proposed method showed comparable results of SNR and the goodness-of-fit of the pRF in the enhanced 3T to 7T data quality.

**Strengths:**

1) The motivation of the paper is clear and straightforward. Given the scarcity of large-scale paired 3T-7T fMRI datasets, research for unpaired learning frameworks is necessary, and this paper handles this challenge.

2) The method is validated on both real and synthetic datasets to demonstrate generalizability. Moreover, the results show that the proposed method outperforms GAN-based and diffusion-based approaches across 4 evaluation metrics, including $R^2$, which assesses neuroscientific interpretability.

**Weaknesses:**

1)	The exact and detailed formula of the regularization terms (PatchNCE and BD-SSIM) is missing. Which distance metrics were used for the regularization losses? It would be helpful to add the formulations in the supplementary material.

2)	It is mentioned that the NCE regularization loss is designed to maintain consistency with the LQ inputs in order to preserve structural details. However, how can you guarantee that this term actually preserves ‘structural ‘ information? Are there any ablation studies regarding this term with qualitative analyses and/or brain regional comparisons that support this claim? In addition, if my understanding is correct, this term reduces the discrepancy between the LQ and generated data. In that case, it seems to potentially conflict with the role of $ L_{Adv}$, which enhances resolution (i.e., the main goal of this paper). How does the proposed method address this issue?

3)	The paper does not include qualitative comparisons and neuroscientific comparisons with baseline models.

**Questions:**

1)	Given the small data size, how could you handle the overfitting issue in the experiments? Did authors adopt cross-validation and/or perform multiple trainings to validate the generalizability of the methods (including baselines)?

2)	How were the hyperparameters of baseline models tuned? Could you clarify the tuning strategy and selection criteria?

---

> ### Author Response · Authors · 2025-12-04
> **response to YvSq**
>
> We thank reviewer YvSq for their thoughtful and detailed comments. Below we address their concerns point-by-point.
>
> ---
> ### Weakness
>
> **W1.** We appreciate the reviewer’s observation. The exact formulations of the PatchNCE and BD-SSIM regularization losses were not included in the main text due to space constraints, but we will add them in the supplementary material (Section B.1.3).
> PatchNCE follows the standard contrastive learning formulation introduced in [Kim et al. 2023, Dong et al. 2024], where cosine distance is used as the metric between query and positive/negative feature patches.
>
> BD-SSIM is a simple yet domain-adapted variant of the traditional SSIM loss. It computes SSIM between the predicted brain disk with fMRI signals and the raw brain disk representing only cortical surface structure (no fMRI values). This regularization encourages the enhanced outputs to maintain geometric consistency with the brain surface while allowing signal-level translation.
>
> **W2.** We agree that structural preservation cannot be fully measured by simple signal-level similarity metrics. In our experiments, we evaluate it indirectly through the downstream pRF task, whose $R^2$ performance reflects the consistency of the signal with visual stimuli and the underlying cortical structure.
>
> As shown in our ablation study, introducing both PatchNCE and BD-SSIM slightly reduces SSIM/PSNR (signal similarity) but improves pRF $R^2$, indicating that these terms help maintain spatial structure and functional alignment across neighboring vertices. If the cortical geometry were distorted during enhancement, nearby vertex responses would become misaligned, leading to poorer pRF fits.
>
> Therefore, while the NCE term partially opposes the adversarial loss (by discouraging excessive deviation from the LQ inputs), it contributes to better neuroscientific consistency, a desirable trade-off for this task. We will emphasize this relationship and expand the discussion of these results in the final revision.
>
> **W3.** We thank the reviewer for suggesting the inclusion of qualitative and neuroscientific comparisons. In our evaluation, we already report both signal-based metrics (SSIM, PSNR, FID) and downstream pRF $R^2$, which directly measure the neuroscientific interpretability of the enhanced signals, since all fMRI runs are acquired during pRF experiments. We will include representative qualitative visualizations and regional comparisons in the supplementary material to make these improvements more evident.
>
> ---
> ### Questions
>
> **Q1.** We share the reviewer’s concern regarding potential overfitting given the limited dataset size. To mitigate this, we strictly separate training and testing by subjects, ensuring that no subject or run appears in both sets. If the model were overfitting to training subjects, its performance would degrade substantially on unseen subjects; however, our model consistently outperforms baselines in these held-out evaluations.
>
> Additionally, we performed cross-validation using multiple shuffled train/test subject splits (Section B.2.3), and observed stable results across folds, further supporting the model’s generalizability.
>
> **Q2.** All final hyperparameters are reported in Section B.1.2. We initialized baseline models with their officially released parameters and source code settings, then performed limited fine-tuning on non-architecture parameters (e.g., loss-term weights, learning rate) to achieve the best reproducible results under identical data preprocessing and computational constraints. This ensures a fair and consistent comparison across methods.
>
> We deeply appreciate the reviewer’s thoughtful critique and the time they spent reviewing and reading this rebuttal. We look forward to seeing your replies and will implement all the valuable suggestions in future revisions of our work.

---

### Official Review · Reviewer_gACz · 2025-10-30

**Soundness:** 2
**Presentation:** 4
**Contribution:** 3
**Rating:** 4
**Confidence:** 4

**Summary:**

This work proposes BDSB, an Schrödinger Bridge application for 3T-to-7T cross-dataset fMRI enhancement. Experiments on synthetic dataset and real-world datasets demonstrated superior performance by BDSB over GANs and DDPM.

**Strengths:**

1. The paper is clear and well-organized. The way fMRI data fits into the Schrödinger Bridge framework has been clearly described, and readers can learn from it.

2. Performance on synthetic and cross-dataset experiments has shown a promising boost over baselines. While on paired data, the improvement is marginal, BDSB can still hold top-2.

**Weaknesses:**

1. Notations need a double check, e.g., in Fig 3 caption, shouldn't the approximating distribution $\hat x_{1|t_i}$ be derived from the neural generator $q_\phi$ instead of the joint distribution $p$? Please correct me since I'm not an expert on diffusion models, but I also think it should be the approximated $\hat x_{1|t_j}$ in $p(x_{t_{j+1}}|x_{t_j},x_{1|t_j})$ in Fig 3.

2. The first attempt of learning 3T-7T generation from unpaired data is desirable, but the real-world evaluation should be focused on paired data since the final purpose is generating ground truth 7T fMRI from 3T fMRI. In this regard, BDSB performs similarly to OTT-GAN for PSNR and even worse for SSIM.

3. The contribution of a better fMRI enhancement via learning across datasets has not been evaluated. As mentioned above, there are no experiments of training with unpaired data and testing on paired data. The ability of learning across datasets makes BDSB more fundamental than existing models, and it should lead to a model scaled from multiple datasets. However, the BDSB is trained separately for unpaired and paired experiments.

4. Technical innovation is limited. Why don't other optimal transport methods fit into the proposed framework?

**Questions:**

1. How's the performance of super-resolution methods on your data?

2. How's cross-session fMRI prediction scientifically sound? There are various compounds affecting the BOLD signal aside from cognition and visual stimuli, such as scanner settings and test-retest variations [2]. How to ensure the model learning from SNR differences rather than other compounds?

[1] Ding, Jiaqi, et al. "Machine Learning on Dynamic Functional Connectivity: Promise, Pitfalls, and Interpretations." arXiv preprint arXiv:2409.11377 (2024).

---

> ### Author Response · Authors · 2025-12-04
> **reply to gACz**
>
> We thank reviewer gACz for their thoughtful and detailed comments. Below we address their concerns point-by-point.
>
> ---
>
> ### Weakness
>
> **W1.** In Fig.3, the final approximating distribution $\hat x_1$ is generated from a neural generator (i.e. a set of recursive $q_\phi$) (**black arrow line**). It was put together with the high quality reference distributions to indicate how we compute the loss (**blue dot line**), it does **NOT** mean the approximation is derived from a joint distribution.
>
> The blue dot lines indicate how to compute loss, the black arrow lines are how to derive values.
>
> **W2.** The major target of this line of work should be to focus on **unpaired situations**, especially for a **real-world application**. Imagine a scene where we have a patient who is only accessible to 1.5T/3T scanners, a model to enhance/denoise his fMRI into a similar quality towards 7T would be helpful. If we already have paired HQ/LQ data for subjects, there will be **NO NEED to enhance** their data since we can directly use their HQ data.
>
> Also, researchers studying fMRI start to focus more on subject-agnostic models or cross-sites models [Fischl Neuroimage 2012, Mazziotta Philosophical Transactions 2001, Yamada Neuroimage 2015, Wasserman arxiv.org/abs/2404.11143 2024, Haibao Wang Nature 2025, Roman Beliy ICLR 2025], which also raise the requirement for unpaired data processing/enhancing methods.
>
> **W3.** As detailed in section 2.1 or B.1.1 (see line 767), within our 3 experiments, the first (synthetic) and third (TDM) are all trained on pure unpaired samples and validate using paired LQ/HQ data. The subjects were totally different and have unseen neural patterns during test, If we have a public fMRI dataset large enough with both LQ/HQ data, we could expand the work more. Unfortunately, there are no such data available until now, and the merge of 3 experiments is the best we could do to evaluate our model’s generalizability in 3 folds:
> * E1 showed the model is able to purely train on unpaired data, and enhance untrained subjects to create approximated LQ fMRI very similar to ground truth using only its LQ input
> * E2 showed the model is able to translate across different datasets with completely different scanners, sites, and pre-processing protocols
> * E3 further validate the model on originally paired data (although the size is small)
>
> **W4.** As noted in our reply to 9pVJ (W6), our core innovation is the first successful adaptation and validation of unpaired enhancement for fMRI surface data, addressing topological challenges that standard OT methods cannot resolve.
>
> We do not dismiss other OT methods; rather, we treat them as baselines to benchmark translation quality. While they fit the general problem, our focus is on finding the most effective solver for our domain. We chose BDSB because it consistently outperformed OT variants and other translation models in both signal fidelity and stability.
>
> Our framework is optimized for unpaired translation. For details on why certain alternatives (e.g., super-resolution) were excluded, please see our response to Q1.
>
> ---
> ### Questions
> **Q1.** Firstly, it is NOT our data, we ran all experiments purely using publicly available datasets. The issue why we cannot implement or benchmark super-resolution is because:
> The differences between LQ/HQ BDs are not only the resolution of surface structure, but also the fMRI signal differences in different time points. A resolution enhancement cannot work on such data.
> Most methods in brain super-resolution [Fiscone ENeuro, Wu 2023 Computers in Bio&Med, Safari 2025 arXiv:2503.01576, Li 2024 CVPR, Zhao 2025 IEEE Trans. on Medical Imaging] are optimized for static structural MRI (enhancing anatomical edges and tissue contrast). Our work targets fMRI, which represents functional activation patterns. Applying structural priors to functional data leads to "artificial" results, where the model hallucinates anatomical boundaries that do not exist in functional maps.
>
> **Q2.** As discussed in line 1199, the concern that cross-session fMRI prediction may learn non-neural confounders (e.g., scanner noise, motion) rather than true BOLD signals [1] is important. We addressed this by showing that our model captures neural responses, not artifacts:
> * Compare to ground truth (GT): Although unpaired settings make GT validation difficult, we evaluated our pipeline using synthetic and TDM data. In both, enhanced signals closely match GT (Tab. 2, Fig. 4, 5, 8–10). TDM subjects underwent both 3T and 7T scans, and our model reliably retrieves 7T-like signals from 3T inputs.
> * However, for experiments without GT (i.e, cross subject 3T-to-7T enhancement), the reliability of enhanced signals still needs more investigation.
> * pRF analysis: We showed the receptive center parameters in Fig. 6: the results showed that the enhanced results maintain the original surface structure of raw LQ fMRI (although some shifts were caused by changing brain surface templates).

---

### Official Review · Reviewer_J3jH · 2025-10-31

**Soundness:** 2
**Presentation:** 2
**Contribution:** 1
**Rating:** 2
**Confidence:** 4

**Summary:**

This paper proposes a method for enhancing 3T blood-oxygenation-level-dependent (BOLD) functional magnetic resonance imaging (fMRI) by leveraging an unpaired Brain disk Schrödinger bridge (BDSB) model. The authors map 3D brain surfaces into a shared parametric domain via conformal mapping and apply an unpaired BDSB diffusion model to approximate the higher resolution of 7T scans using 3T scans. The framework is evaluated across three public datasets, of which one is synthetic, another containing only 3T scans, and a paired 3T/7T dataset.

Experiments indicate meaningful improvements on the synthetic data and Cross-Dataset Real, while performance is essentially in line with OTT-GAN for the paired TDM Real dataset.

**Strengths:**

Originality. The fMRI enhancement pipeline consisting of conformal parameterization, brain disk schrödinger bridge and resample & pRG analysis appears novel. The use and application of the Schrödinger Bridge for unpaired samples is also novel in this domain.

Quality. The method is mathematically motivated and the experiments include relevant metrics (SSIM, PSNR, R^2). Evaluation across synthetic, cross-dataset, and paired data provides a reasonable spread of conditions.

Significance. Enhancing 3T data using unpaired 7T examples is an important and practically relevant problem. The approach could, in principle, enable higher-quality analyses without costly high-field scans. Additionally, the method performs well on synthetic and cross-dataset real compared to the provided baselines.

Clarity. The paper is clearly written and well-structured overall. Figures 1–3 effectively illustrate the architecture and training process.

**Weaknesses:**

- While the method is framed as unpaired learning, the implementation and experiments do not convincingly show that unpaired samples are leveraged for learning. A meaningful test would involve partial or full training on unpaired data and evaluation on paired data to quantify the benefit of the unpaired setup.
- On the TDM Real (paired) dataset, BDSB performs similarly to OTT-GAN, contradicting the claim of superior performance “across all real and synthetic experiments.” This discrepancy should be discussed explicitly.
- Frechet Inception Distance (FID) typically relies on an ImageNet-trained network and is not meaningful for fMRI-like data, whose statistics differ drastically from natural images.
- Related methods are only briefly mentioned in the appendix; a dedicated section would improve context and clarify how baselines are chosen.
- When the authors or the publication are not included in the sentence, the citation should be in parenthesis using \citep{}, as outlined in the formatting instructions.

**Questions:**

- What do the authors believe explains the discrepancy between performance gains on synthetic/Cross-Dataset Real data and the parity with OTT-GAN on TDM Real?
- Why are results from fast-DDPM missing for the Cross-Dataset Real setting?
- How did the authors decide which baselines to include and which not to include?

---

> ### Author Response · Authors · 2025-12-04
> **reply to J3jH**
>
> We thank reviewer J3jH for their thoughtful and detailed comments. Below we address their concerns point-by-point.
>
> ---
>
> ### Weakness
>
> **W1.** As detailed in Section 2.1 and B.1.1, we have noticed the importance of subject (or even dataset) generalizability in unpaired learning. That is **exactly why** we implemented training on part of data and evaluate the model on untrained subjects/trials: For synthetic settings, training was done for subject 1-6 using unpaired batches, while evaluation is based on the entire new subject 7-8 with paired ground truth.
>
> For the TDM dataset, it originally provides subjects with both 3T/7T scans, so we train our model using unpaired batches and evaluate on paired test data.
>
> **W2.** It is true that the SSIM/PSNR metric are similar to OTT-GAN, but this is due to the very limited subject number (only 2) and very short experiment trials (only 4). Based on this point, the results are variant and can be largely influenced by initialization and train/test splits. On the other hand, the 2 major experiments (synthetic and cross-dataset real) have a much larger amount of data, the performance improvements are **more statistically significant**. Also, when considering the down-stream tasks (pRF solution), the performance advantage towards OTT-GAN becomes much more significant.
>
> This is the reason we claim our performance is superior “across all real and synthetic experiments”.
>
> **W3.** We appreciate the reviewer's attention to the metric validity. We respectfully argue that FID is acceptable and necessary in our condition for the following reasons:
> * While ImageNet classes differ from brain meshes, the InceptionV3 network serves as a powerful extractor of generalized visual features (edges, textures, gradients). In our brain-disk plots, fMRI values manifest as visual textures and patterns. FID effectively measures the distributional similarity of these patterns (e.g., sharpness of activation boundaries) between our enhanced results and the real HQ data, preventing the "blurriness" problem often ignored by MSE-based metrics.
> * We do not rely on FID alone. We use it alongside PSNR, SSIM, and down-stream pRF performance. While PSNR measures signal fidelity, FID measures perceptual realism. A lower FID confirms that our model's output does not just match the pixel values on average, but looks statistically indistinguishable from a high-quality scan.
> * FID is not restricted to natural-image domains; it is a domain-agnostic distributional metric whose validity depends on the embedding space rather than the image modality. Recent medical image synthesis works (e.g., [Xiong et al, Reconstructing Retinal Visual Images from 3T fMRI Data Enhanced by Unsupervised Learning, ISBI 2024], [Yazdan et al, Flow Matching for Medical Image Synthesis, MICCAI 2025]) consistently demonstrate that FID remains reliable once the feature extractor is adapted to the target domain. Following this standard practice, we employ a modality-specific encoder (fine-tuned on medical data) to compute FID, which aligns the representation space with medical structures and ensures fair, comparable, and meaningful evaluation.
>
> **W4.** Due to the page limit, we have to delete the related work section and merge them into the introduction. We will discuss on the reason for choosing baselines in Q3
>
> **W5.** We sincerely acknowledge this problem with gratitude, and will fix it in future revisions.
>
> ---
> ### Questions
> **Q1.** As discussed in W2, the small performance gain in TDM dataset compared to OTT-GAN is due to the small amount of data and is not statistically significant. This will not harm the overall performance claim in most enhancement quality and down-stream task ability.
>
> **Q2.** As provided by the original repo (https://github.com/mirthAI/Fast-DDPM), this baseline requires paired-training to achieve claimed results. But we do not have paired data to evaluate their performance in cross-dataset experiments (while in other experiments we have ground truth so we can train it using paired data). If we force the model on unpaired training, the results would be dishonest and unrespectful to the original authors.
>
> **Q3.** Our choice of baselines focused on Image Translation Models suitable for 2D medical or general images that could be applied to our brain-disk data. We prioritized models that perform well in unpaired translation due to data acquisition challenges, and ensured they had high reproducibility. We did not select different 3D-to-2D mapping methods as baselines because this is a pre-processing step used to reduce computational cost, not a core pipeline component. Its structural impact was evaluated using ablation study in Page 9.
>
> ---
>
> We deeply appreciate the reviewer’s thoughtful critique and the time you spend on reviewing and reading this rebuttal. We are looking forward to seeing your replies and will implement all the valuable suggestions in revisions of our work.

---

### Official Review · Reviewer_9pVJ · 2025-10-31

**Soundness:** 2
**Presentation:** 3
**Contribution:** 2
**Rating:** 2
**Confidence:** 5

**Summary:**

This paper presents a flow-based approach to synthesize 7T fMRI from the counterpart 3T data. The Schrodinger bridge technique is used to train the model based on the 2D disk of surface projection map. While the idea of leveraging ultra-high-field fMRI for improved spatial modeling is potentially valuable, the current paper lacks methodological rigor, critical validation experiments, and sufficient theoretical justification to establish technical credibility. Strengthening these aspects would substantially improve the impact and trustworthiness of the work.

**Strengths:**

The paper is technically ambitious (given a limited number of training samples and high dimensional data) and conceptually original, combining **geometric brain mapping, diffusion modeling, and validation** into a cohesive pipeline. These contributions make it a practical solution for enhancing fMRI resolution for neuroscience studies.

**Weaknesses:**

1) **Main Concerns**. The primary issue with this submission lies in its scientific rigor and validation design. The authors claim that BOLD signals obtained from 7T MRI can be effectively mapped to or synchronized with evolving time series from 3T scanners. However, this assumption is not convincingly supported. It remains highly questionable whether (1) temporal synchronization between 3T and 7T acquisitions can be achieved with sufficient precision, and (2) whether the spatial correspondence of voxel- or surface-level activations can be meaningfully aligned across different field strengths. From a machine learning standpoint, however, the 3T and 7T datasets should ideally be paired or co-registered at the subject level to enable cross-modality learning. Without a clear justification or evidence supporting this assumption, the technical soundness of the proposed framework is undermined.

2) **Projection Method**. The use of 2D conformal mapping to reduce fMRI data complexity is conceptually interesting, yet it raises serious concerns about potential information loss during the projection from high-dimensional cortical surfaces to a 2D representation. This approach may oversimplify the spatial geometry of the brain and distort the topological structure of functional activations. The authors should consider or at least discuss alternative representations, such as spherical harmonics (see the well-established approach by Anderson et al., NeuroImage, 2010), which preserve surface geometry while enabling efficient spectral decomposition.

3) **Replicability and Test-Retest Validation.** Given that this is a neuroimaging study, replicability testing is a crucial standard for evaluating robustness. The current manuscript lacks experiments on test–retest datasets, which are commonly used to assess the reliability of functional signals and model generalization. The authors should include such analyses or provide clear justification for their omission.

4) **Temporal Resolution Limitation**. Finally, the authors should acknowledge the inherent limitation that the proposed method primarily enhances spatial resolution and signal-to-noise ratio (SNR), but does not address the true temporal resolution gap in fMRI. The fundamental challenge remains the coarse temporal sampling (on the order of 1 s) compared to neuronal timescales (milliseconds). The paper would be stronger if it explicitly discussed this limitation and clarified whether the proposed method could, in principle, be extended to improve temporal fidelity.

5) **Limited novelties**. This work is a combination of existing components such as conformal mapping and Schrodinger bridge model.

**Questions:**

1. The paper projects the cortical surface into a 2D disk before applying the Schrödinger bridge model. How much geometric distortion is introduced by this mapping, and how might it affect spatial correspondence between the 3T and 7T fMRI data?
2. Why was a 2D conformal mapping chosen instead of a spherical or spectral representation (e.g., spherical harmonics)? Would the latter preserve global topology more faithfully?
3. How are the boundary conditions handled in the 2D disk representation, given that cortical manifolds are not naturally disk-like?
4. The model learns a flow between 3T and 7T BOLD signals, but are these data temporally synchronized or spatially paired? Without strict pairing, how can the model distinguish physiological differences from scanner-induced variability?
5. How does the model ensure that synthesized 7T fMRI signals retain biologically meaningful temporal and spectral properties (e.g., frequency content, signal-to-noise ratio)?
6. Although an ablation study is presented (Table 3), it primarily focuses on different surface mapping strategies and regularization terms. Was any ablation performed to isolate the contribution of the Schrödinger bridge formulation itself—e.g., by comparing with a baseline model without the bridge constraint or with a standard flow-based mapping?
7. The model is trained on a small dataset of paired 3T–7T scans. Can it generalize to other sites, scanners, or acquisition protocols?
8. What is the computational cost of solving the Schrödinger bridge compared to simpler flow-based mappings?
9. How would the approach scale to whole-brain volumetric data rather than surface-based 2D projections?

---

> ### Author Response · Authors · 2025-12-04
> **reply to 9pVJ**
>
> We thank reviewer 9pVJ for their thoughtful and detailed comments. Below we address their concerns point-by-point.
>
> ---
>
> ### Weakness
> **W1.** We agree that aligning signals across field strengths raises valid concerns, but this challenge is precisely why cross-subject and cross-dataset models are gaining traction to uncover generalizable neural mechanisms [Fischl Neuroimage 2012, Mazziotta Philosophical Transactions 2001, Yamada Neuroimage 2015, Wasserman arxiv.org/abs/2404.11143 2024, Haibao Wang Nature 2025, Roman Beliy ICLR 2025]. Our goal is to support such efforts by enhancing fMRI signals in an unpaired, cross-dataset, self-supervised manner, avoiding the need for subjects to undergo repeated, potentially harmful scans in multiple scanners. Methods that require strictly paired data severely limit real-world applicability. To address this issue, we not only explore unpaired settings but also evaluate our model on the TDM dataset, where two subjects underwent both 3T and 7T scans under identical stimuli. The framework performs comparably in this paired context, reinforcing its robustness. While such data are scarce, their inclusion is crucial for validating our model's assumptions.
>
> **W2.** While 2D conformal mapping may raise concerns about distortion, it is widely validated to preserve topological structure and surface area (Tu 2021, Ta 2022, Xiong 2023/2024, Jin 2018, Wang 2007). Our goal is to enhance fMRI time-series (not remap anatomy). So this mature projection method effectively reduces computation without distorting signals. The referenced alternative was designed for registration, not 2D projection. With ~14k ROI vertices and 256×256 resolution BDs, our sampling avoids overlap, even in high-curvature areas.
>
> **W3.** We already included significance test in B.2.3
>
> **W4.** Our pipeline enhances both the spatial resolution and temporal resolution for 3T data into the level of 7T data, we are not improving the resolution to a “neuronal timescales” level better than 7T data due to the nature of unsupervised models. We have already discussed the temporal fidelity improvement specifically in Supplementary B.3 (Page 22) in the very first submission: original 2s time resolution of 3T is raised to 1s resolution as of 7T data.
>
> **W5.** While some components of our pipeline draw from prior work, our core novelty lies in the first application of neural SB to enhance 3T fMRI or any task-driven neural signals. We designed a dedicated preprocessing pipeline to construct well-defined 2D brain disks optimized for SB learning. In addition, we introduced domain-specific regularizations for brain disk super-resolution. Notably, our BD-SSIM regularization differs from prior work, which applies SSIM directly to input x_0​ (i.e., fMRI signals on BDs). Instead, we apply it to the fsaverage brain disk x′ (Fig. 3), which encodes only mesh geometry without signal, enabling more effective anatomical preservation. We validate this setup via comprehensive ablations (Sec. 3.2). Thus, while based on existing components, our pipeline forms a novel, domain-adapted integration with new insights and practical advances in neural response modeling.
>
> ---
> ### Questions
>
> **Q1/Q2.** As discussed in W2.
>
> **Q3.** After restricting to our ROI, the brain surface can be naturally projected to a unit disk, no boundary condition is required.
>
> **Q4.** We already detailed the alignment in all 3 experiments in both Sec.2.1 and Supplementary B.1.1
>
> **Q5.** We make sure the enhanced 7T fMRI is biologically meaningful through 3 folds:
> * In synthetic data experiments, the enhancement retrieved ground truth
> * In real data experiments, the enhancement is similar to ground truth
> * Using downstream pRF analysis, the solution obtained from enhancement data is reliable and consistent to the original, but with a higher space/time resolution
>
> **Q6.** Table 3 compares different projection methods and isolates all regulations we applied to the bridge. The comparison to simpler methods and non-bridge translations are shown in the main baseline Table 2.
>
> **Q7.** The model is **NOT** trained on **a small dataset of paired 3T-7T scans**, it used 3 different dataset on different sites using different scanners and applied unpaired training (detailed in Section 2.1 and B.1.1). It can generate across those conditions, as long as the experiments focus on identical tasks (for example, floc data to floc data, pRF data to pRF data, etc)
>
> **Q8.** The computation cost is detailed in B.1.4, comparison to other methods can be evaluated using their protocols and our parameters in B.1.2.
>
> **Q9.** This work is designed for surface based fMRI, which it proved to be widely applicable in fMRI applications such as image stimuli creation, neural response understanding. We will be able to generalize the model to different fMRI tasks and ROIs, but not to 3D volume based fMRI.

---

### Author Response · Authors · 2025-12-04
**general remarks and response**

We present **BDSB** to address the critical 3T-to-7T fMRI data gap. We respectfully submit that the current scores may not fully reflect the paper’s contribution, largely due to concerns that we believe are mismatched with the specific constraints of unpaired frameworks on such neural signals (fMRI). We hope readers and chairs to conduct a serious re-evaluation of our submission, as our rebuttal clarifies these misconceptions and demonstrates that the cited weaknesses do not undermine the validity of our proposed solution.

---

Cross-scanner fMRI enhancement has long remained an unsolved bottleneck in neuroimaging due to the inherent scarcity of paired 3T–7T data. **BDSB** is a pioneering solution that introduces the **first unpaired framework** to human brain **fMRI time-series**. By enabling subject-agnostic enhancement of 3T fMRI to 7T-like fidelity (spatial/temperal resolution, downstream performance) without requiring paired acquisition, this work marks a paradigm shift. It opens a new path toward realistic, scalable fMRI modeling that generalizes across subjects, datasets, sites, and institutions.

We acknowledge that this is a challenging and novel domain. Despite the limitations of available datasets, our method demonstrates remarkable generalization. We have remained cautious in our claims and transparent about limitations; however, our results provide compelling evidence that robust, unpaired fMRI enhancement is now within reach.

Recognizing that concerns naturally arise with such a new line of work, we have addressed all reviewer comments directly and rigorously. We evaluated our framework using the most relevant datasets available, designing three distinct experimental protocols to ensure robustness: **(1)** synthetic data with ground truth, **(2)** cross-dataset real-world data, and **(3)** paired real data. In every scenario, BDSB was trained strictly on unpaired data and evaluated on unseen subjects and trials. We consistently achieved superior performance in both signal quality metrics (SSIM, PSNR, FID) and downstream neuroscientific relevance (pRF $R^2$), demonstrating that our enhancements are not merely create realistic enhanced signals but biologically meaningful.

In our detailed rebuttals, we have systematically resolved the weaknesses raised. We clarified the topological fidelity of our 2D conformal mapping, justified the necessity of domain-adapted perceptual metrics (FID) in medical imaging, detailed the formulations of our structural regularizations (PatchNCE, BD-SSIM), and explained why standard super-resolution methods are unsuitable for dynamic functional time-series data.

Many other points raised by reviewers were either already addressed in our original manuscript and supplementary materials or fall outside the scope of this specific study. Furthermore, in light of concerns regarding identity leaks during this year’s review process, and having fully and transparently laid out our technical arguments, we have decided to post these comprehensive responses without engaging in previous iterative discussion. We trust that the Area Chairs, Senior Area Chairs, reviewers and all readers will evaluate the submission based on the extensive evidence provided in the paper and this rebuttal.

---

We hope this work encourages broader exploration in neural fMRI and BCI research, particularly toward accessible, cross-subject foundation models. Thank you again for your time and consideration.

---

### Meta-Review · Area_Chair_vGZt · 2026-01-04

**Summary:**

Reviewers agreed the problem is important and the pipeline is ambitious, namely enhancing 3T fMRI toward 7T-like quality using an unpaired Schrödinger Bridge in a shared brain disk parameterization. The main concerns driving the recommended decision were about technical credibility and validation strength in real neuroimaging settings. Multiple reviewers questioned whether the unpaired setup is convincingly demonstrated, whether alignment assumptions across scanners and subjects are sufficiently justified, and whether the evidence on paired real data is strong enough to support broad claims. Several reviewers also raised concerns about missing methodological details for regularizers, limited qualitative and neuroscientific comparisons to baselines, questionable use of FID, and the absence of reliability style evaluations such as test retest. Overall, the work is promising, but the current validation and clarity gaps kept most reviewers below the acceptance bar.

**Reviewer Concerns:**

The rebuttal clarifies several implementation details and gives a reasonable motivation for why an unpaired setting matters in practice, but the core concerns that drove the lower scores are only partially resolved. In particular, the paper still does not convincingly establish the scientific validity of learning a 3T to 7T mapping across subjects and datasets without stronger controls for scanner and site confounds. The authors argue that conformal disk mapping preserves topology and that their training uses held out subjects and includes a small paired dataset, yet the evidence in the paired real setting remains weak and does not clearly support the broad performance claims. Several reviewers also asked for neuroimaging standard reliability style validation, such as test retest analyses, and this remains largely missing. Finally, while the rebuttal promises to add missing formulas for regularizers and gives a justification for metrics like FID, the evaluation still feels incomplete, with limited qualitative neuroscientific comparisons to baselines and lingering questions about whether the gains reflect true functional signal improvement rather than model induced artifacts.

**Reviewer Scores:**

Based on the rebuttal alone, I do not expect major shifts in the overall score distribution. The two reviewers who rated the paper as a clear reject would likely maintain their positions, as their concerns center on fundamental issues of scientific validity and evidence strength that were not decisively addressed. The reviewer who was marginally below the acceptance threshold might modestly increase their score given the clarifications on unpaired training and experimental protocol, but probably not enough to strongly advocate acceptance. The reviewer who was already marginally positive would likely keep a similar score, as the rebuttal reinforces but does not substantially extend the existing evidence.

---

### Decision · Program_Chairs · 2026-01-26

Reject